# ADAGRAD UNDER ANISOTROPIC SMOOTHNESS

**Yuxing Liu**[*]     **Rui Pan**[*]     **Tong Zhang**
University of Illinois Urbana-Champaign
{yuxing6,ruip4,tozhang}@illinois.edu

## ABSTRACT

Adaptive gradient methods have been widely adopted in training large-scale deep neural networks, especially large foundation models. Despite the huge success in practice, their theoretical advantages over classical gradient methods with uniform step sizes across all coordinates (e.g. SGD) have not been fully understood, especially in the large batch-size setting commonly used in practice. This is because the only theoretical result that can demonstrate this benefit was obtained in the original paper of Adagrad for convex nonsmooth objective functions, which is insufficient for large batch algorithms. In this work, we attempt to resolve this gap between theory and practice by proposing a novel anisotropic generalized smoothness assumption and providing corresponding analysis of Adagrad. It is shown that under anisotropic smoothness and noise conditions, AdaGrad can achieve faster convergence guarantees in terms of better dimensional dependence than algorithms with uniform step sizes across all coordinates. Experiments in logistic regression and instruction following fine-tuning tasks provide strong evidence to support our novel assumption and theoretical analysis.

## 1 INTRODUCTION

To solve the stochastic optimization problem

$$\min_{\mathbf{w} \in \mathbb{R}^d} f(\mathbf{w}) \triangleq \mathbb{E}_\xi[f(\mathbf{w};\xi)], \tag{1}$$

adaptive gradient methods (Duchi et al., 2011; Zeiler, 2012; Tieleman et al., 2012; Kingma & Ba, 2014; Loshchilov & Hutter, 2017) are among the most popular methods. These methods have gained incredible importance from their superior efficiency, especially in training large foundation models, where large batch sizes are commonly employed. One of the most important features of adaptive gradient methods is the coordinate-wise anisotropic step size. Take Adagrad (Duchi et al., 2011; Streeter & McMahan, 2010), the first adaptive gradient method as an example, which writes

$$\mathbf{w}_{t+1} = \mathbf{w}_t - \eta_t \mathbf{\Lambda}_t^{-1} \mathbf{g}_t,$$

where $\mathbf{g}_t$ notes the stochastic gradient estimation obtained at $\mathbf{w}_t$ and $\mathbf{\Lambda}_t \neq \mathbf{I}_d$ is a diagonal matrix, representing the square root of a coordinate-wise summation of former gradient estimations.

Despite the huge success in practice, as also noted in Li et al. (2021); Kunstner et al. (2023); Li et al. (2024), the theoretical understanding of when and why adaptive gradient methods enjoy acceleration over classical gradient algorithms with uniform step size across all coordinates such as SGD, is still limited. On the theory side, the original Adagrad paper Duchi et al. (2011); Streeter & McMahan (2010) shows the superiority of Adagrad over SGD in the convex non-smooth scheme, suggesting that if the gradients are sparse and the predictor is limited in an appropriate convex set, Adagrad can converge faster in terms of better dimensional dependence. However, as the convergence rates in nonsmooth settings rely on the scale of stochastic gradients, their results can be insufficient for the smooth and large-batch training scheme, which is a realistic setting gaining extensive focus. This is because the scale of stochastic gradients does not decrease linearly as the batch size $M$ increases. Therefore, if we fix the stochastic gradient computation number $N = MT$, where $M$ is the batch size and $T$ is the total iteration number, such that increasing $M$ leads to a linear decrease in $T$,

---

[*]Equal contribution.

the original convergence rate $\mathcal{O}(1/\sqrt{T})$ may be unsatisfying. Fine analysis in smooth settings can solve this problem as the obtained convergence results in this case depend on the gradient variance, which decreases linearly as the batch size $M$ increases. Many theoretical papers have also conducted analysis of popular adaptive gradient methods (e.g. Adagrad, Adam) in smooth settings under smoothness and noise assumptions with respect to $\|\cdot\|_2$. However, to the best of our knowledge, their proven results have no better or even worse dimensional dependence than the standard convergence results of SGD in the same settings. The existence of this gap between theory and practice makes us wonder: *Can we obtain better theoretical guarantees of adaptive gradient methods to explain their practical success, especially in large batch settings?*

To answer this question, it would be a good starting point to revisit the well-known insight of adaptive gradient methods. Intuitively, Adagrad shines when the problem is highly imbalanced, i.e., coordinates of the gradients have very different magnitudes. It schedules a larger learning rate (compared to SGD) for coordinates with small gradients and thus converges faster. This implies that the performance of adaptive gradient methods relies largely on the anisotropic structure of the problem. However, existing standard assumptions fail to describe this property. Take the standard smoothness (Nesterov et al., 2018) as an example, which assumes a constant $L > 0$ such that $-L\mathbf{I}_d \preceq \nabla^2 f(\mathbf{w}) \preceq L\mathbf{I}_d$ for all $\mathbf{w}$. It is evident that this assumption is coordinate-wise equivalent and cannot reflect the imbalance between coordinates, thus the benefits of adaptive gradient methods are hidden, as in the results of Vaswani et al. (2020); Défossez et al. (2020); Wang et al. (2023).

To better explore the provable benefits of adaptive gradient methods, it is necessary to employ appropriate assumptions that can better describe the structure of models. In this paper, we consider the anisotropic smoothness and noise assumptions. Based on these assumptions, we give novel convergence analysis of Adagrad and show that Adagrad can adapt well to the problem's anisotropic nature. To extend our results to a more general setting, we additionally introduce a novel generalized anisotropic smoothness assumption and provide corresponding analysis of AdaGrad. By comparisons between the convergence results of Adagrad and gradient methods with step sizes across all coordinates (SGD and AdaGrad-Norm (Streeter & McMahan, 2010) as two representatives), we justify the power of adaptive gradient methods when the problem has highly anisotropic nature. [1]

Our contributions are summarized as follows:

1. We present a fine-grained analysis of Adagrad under anisotropic smoothness and noise assumptions, leading to novel theoretical convergence guarantees for Adagrad. We further introduce a generalized form of anisotropic smoothness, extending these results to more practical settings.

2. We discuss how the convergence results indicate the potential benefits of AdaGrad compared to algorithms with coordinate-wisely uniform step sizes such as SGD and AdaGrad-Norm.

3. Experiments on logistic regressions on real-world datasets and instruction-following fine-tuning with GPT-2 provide concrete empirical evidence to support our claims.

## 2 RELATED WORK

**Adaptive gradient methods.** Adaptive gradient methods are popular optimizers for training neural networks (Choi et al., 2019; Vani & Rao, 2019). Among them, Adagrad (Duchi et al., 2011; Streeter & McMahan, 2010) is considered to be the first adaptive gradient method in this branch, which was originally proposed to solve non-smooth online convex optimization problems. Ever since its first appearance, numerous adaptive gradient methods have emerged, such as RMSProp (Tieleman et al., 2012), AdaDelta (Zeiler, 2012), Adam (Kingma & Ba, 2014), SC-Adagrad (Mukkamala & Hein, 2017), AdamW (Loshchilov & Hutter, 2017), WNGrad (Wu et al., 2018), AMSGrad (Reddi et al., 2019), SAdam (Wang et al., 2019) to name a few. They have revolutionized the field of deep neural network training, and are still widely adopted in the literature of large language models (Radford et al., 2019; Touvron et al., 2023; Touvron et al.).

---

[1]Note that the comparison between upper bounds is commonly adopted in past literature, e.g. (Bernstein et al., 2018a; Allen-Zhu, 2018), though theoretically it only suggests the worst-case performance comparison of algorithms.

**Convergence results of SGD.** Stochastic gradient descent (SGD), a popular optimizer for many real-world tasks, has been extensively studied in the literature (Robbins & Monro, 1951; Nemirovski & Yudin, 1978; Nemirovskij & Yudin, 1983; Nemirovski et al., 2009; Hazan et al., 2007; Rakhlin et al., 2011; Shamir & Zhang, 2013). In the smooth stochastic optimization scheme, Moulines & Bach (2011) and Ghadimi & Lan (2013) gave analysis of SGD under standard assumptions in the convex and nonconvex settings separately. More recently, Zhang et al. (2019) introduced the $(L_0, L_1)$-smoothness to relax the standard smoothness assumption and studied the convergence of SGD and clipped SGD under this assumption. A line of following work appeared after that (Zhang et al., 2020a; Chen et al., 2020; Gorbunov et al., 2020; Qian et al., 2021; Koloskova et al., 2023).

**Convergence results of Adagrad.** As the pioneering work of Adagrad, Duchi et al. (2011) provided an analysis for Adagrad's convergence guarantees in online convex optimization settings, showing acceleration over SGD when the gradients are sparse. However, the presented analysis is only for general nonsmooth objectives, which is insufficient to explain Adagrad's effectiveness under large batch settings. After that, Levy et al. (2018); Vaswani et al. (2020) studied the smooth convex convergence but showed no better results than SGD. On another side, the convergence of Adagrad or its close variants in the smooth nonconvex setting has been extensively studied (Li & Orabona, 2019; Défossez et al., 2020; Ward et al., 2020; Faw et al., 2022; 2023; Wang et al., 2023; Kavis et al., 2022; Liu et al., 2023b; Attia & Koren, 2023; Hong & Lin, 2024b). Among them, Ward et al. (2020) obtained the $\mathcal{O}(1/\sqrt{T})$ rate of Adagrad-Norm under global bounded gradient assumption. More recently, Faw et al. (2023); Wang et al. (2023) further improved the result to hold under the $(L_0, L_1)$-smoothness assumptions. There are also extensive studies focused on the convergence of Adam and its close variants, we list some of them here for reference: (Reddi et al., 2019; De et al., 2018; Défossez et al., 2020; Guo et al., 2021; Zhang et al., 2022; Wang et al., 2022; Li et al., 2024; Wang et al., 2024; Hong & Lin, 2024a).

**Theoretical understanding of adaptive gradient methods.** Surprisingly, though the community tends to have a common sense that adaptive gradient methods converge much faster than algorithms with uniform step sizes in specific tasks, theoretical explanations on when and why this happens are relatively rare compared to extensive empirical studies. It is worth pointing out that among all the above-mentioned works on theoretical analysis of adaptive gradient methods, only the initial Adagrad analysis (Duchi et al., 2011) clearly shows this acceleration in terms of possibly lower dimensional dependence in the online convex programming setting. How adaptive gradient methods can help accelerate convergence in smooth or large batch stochastic optimization still remains unclear. We also notice that Cesa-Bianchi et al. (2007); Orabona & Pál (2015; 2018); Zhuang et al. (2022) mentioned the intuition of scale-free algorithms and Zhuang et al. (2022) demonstrates the connection between scale-freeness of adaptive gradient methods and better condition number dependence. More recently, Zhang et al. (2024); Das et al. (2024) also investigated why Adam is effective in certain tasks compared to SGD and gives theoretical results in quadratics settings. However, their results are more intuitive with very restrictive analysis that are insufficient to explain real-world tasks. In another line of work, Bernstein et al. (2018a); Wu et al. (2020); Kunstner et al. (2023); Liu et al. (2023a) suggests a relation between the benefits of adaptive gradient methods and their sign-based nature. These intuitions may shed light on the theoretical understanding of adaptive gradient methods.

**Large batch training.** Large batch training enjoys extensive focus for its practical impact. It has been observed that large batch sizes are beneficial for accelerating large model training in practice (You et al., 2017a;b; 2018; 2019; Pan et al., 2022). Furthermore, large batch training is a valuable acceleration technique in distributed machine learning (Verbraeken et al., 2020) and pretraining (Zhou et al., 2023), where adaptive gradient methods are popular. In particular, it is a common practice in pretraining of large language models to combine large batch sizes with adaptive gradient methods (Radford et al., 2019; Touvron et al., 2023; Touvron et al.).

**Concurrent Work.** In parallel with our work, Maladkar et al. (2024) also investigates the use of anisotropic assumptions to describe the nonconvex convergence of AdaGrad. Though the starting points are similar, there are some notable differences between the two works. First, our result applies to a more general smoothness assumption that can well describe practical neural network training (as in Figure 1) and covers the result in Maladkar et al. (2024). Second, We include a convex part in our paper, which is a simpler case for illustrating the power of anisotropic assumptions. Based on this part and our new assumption and theory, we also did various empirical verifications in the experiment part, which provides strong evidence for our theory. Third, Maladkar et al. (2024) further introduced a

lower bound for gradient $\ell_1$-norm convergence of SGD, which may help a more rigorous comparison between SGD and AdaGrad in the nonconvex case. However, it's not clear whether $\ell_1$-norm is a better convergence measure and it depends on the training curvature. To conclude, both the work contribute to a better theoretical understanding of the benefits of adaptive gradient methods.

## 3 PRELIMINARIES

### 3.1 NOTATIONS

We use $\odot$ to denote the coordinate-wise product of vectors and without leading to confusion, $\sqrt{\cdot}$ is sometimes used to denote the coordinate-wise square root of a vector or diagonal matrix. Let $\mathbf{H} \in \mathbb{R}^{d \times d}$ be a symmetric positive definite matrix, we denote the vector norm induced by $\mathbf{H}$ that $\|\mathbf{w}\|_{\mathbf{H}}^2 \triangleq \mathbf{w}^\top \mathbf{H} \mathbf{w}$. With a slightly abuse of notation, for a vector $\mathbf{h} \in \mathbb{R}_+^d$, we denote $\|\mathbf{w}\|_{\mathbf{h}}^2 = \sum_{j=1}^d \mathbf{h}_j \mathbf{w}_j^2$. For a symmetric positive definite matrix $\mathbf{H} \in \mathbb{R}^{d \times d}$ and convex set $\mathcal{W}$ we introduce the $\mathbf{H}$-based projection operator $\Pi_{\mathcal{W}}^{\mathbf{H}}(\cdot)$ such that $\Pi_{\mathcal{W}}^{\mathbf{H}}(\mathbf{w}) = \operatorname{argmin}_{\mathbf{z} \in \mathcal{W}} \|\mathbf{z} - \mathbf{w}\|_{\mathbf{H}}^2$. As discussed in Hazan et al. (2007), the projection is a convex program and can be solved efficiently.

Let us denote $\nabla_{\mathbf{w}} f(\mathbf{w}; \xi)$ the stochastic gradient oracle at $\mathbf{w}$ and $\mathbf{g}_t$ the gradient estimation employed at $\mathbf{w}_t$. $\mathcal{F}_t \triangleq \sigma(\mathbf{g}_0, \cdots, \mathbf{g}_{t-1})$ stands for the sigma field of the gradient estimators from the first iteration to the $t-1$ iteration. We use $\mathbb{E}[\cdot]$ to denote total expectation over $\mathcal{F}_T$ where $T$ is the maximum iteration number and $\mathbb{E}_t[\cdot]$ as an abbreviation of the conditional expectation $\mathbb{E}[\cdot | \mathcal{F}_t]$.

### 3.2 PROBLEM SETTINGS AND ASSUMPTIONS

We study the stochastic optimization problem (1), where we can only access the stochastic gradient oracle $\nabla f(\mathbf{w}; \xi)$ at $\mathbf{w}$. Throughout this paper, we consider the following assumptions.

**Assumption 3.1** (Convexity). $f(\cdot)$ is convex. For convex cases, we search solution in a closed convex set $\mathcal{W} \subseteq \mathbb{R}^d$ such that there exists at least one optimal solution $\mathbf{w}_* \in \mathcal{W}$ and

$$\max_{\mathbf{w}, \mathbf{w}' \in \mathcal{W}} \|\mathbf{w} - \mathbf{w}'\|_\infty \leq D_\infty \quad \text{and} \quad \max_{\mathbf{w}, \mathbf{w}' \in \mathcal{W}} \|\mathbf{w} - \mathbf{w}'\|_2 \leq D_2. \tag{2}$$

**Assumption 3.2** (Lower bounded). There exists constant $f^*$ such that for $\mathbf{w} \in \mathbb{R}^d$, $f(\mathbf{w}) \geq f^*$.

**Assumption 3.3** (Anisotropic $\mathbf{L}$-smoothness). There exists a positive vector $\mathbf{L} = [L_1, \ldots, L_d] \in \mathbb{R}_+^d$ such that $f(\cdot)$ is $\mathbf{L}$-smooth, namely, for $\mathbf{w}, \mathbf{w}' \in \mathbb{R}^d$,

$$\|\nabla f(\mathbf{w}) - \nabla f(\mathbf{w}')\|_{\mathbf{L}^{-1}} \leq \|\mathbf{w} - \mathbf{w}'\|_{\mathbf{L}}. \tag{3}$$

Assumption 3.1 and 3.2 are standard for convex and nonconvex problems, respectively. Assumption 3.3 is an anisotropic generalization of the smoothness condition, which has also been employed in a line of work on SignSGD (Bernstein et al., 2018a;b). It is also worth pointing out that a similar block Lipschitz assumption is commonly employed in the study of coordinate descent (Wright, 2015; Richtárik & Takáč, 2016). It can be an interesting direction to find out the relations in between. The intuition of Assumption 3.3 can be understood in the following manner. When the loss function $f(\cdot)$ is twice-differentiable, Assumption 3.3 is equivalent to $\nabla^2 f \preceq \operatorname{diag}(\mathbf{L})$ that implies the standard smoothness assumption by $L = \|\mathbf{L}\|_\infty$. When the Hessian of $f(\cdot)$ is imbalanced, namely, coordinates have very different scales, $\mathbf{L}$ can be adapted to this imbalanced distribution and can describe a tighter upper bound of the Hessian of $f(\cdot)$, resulting in $\|\mathbf{L}\|_1 \ll Ld$. This benefit is realistic as the highly imbalanced spectrum distribution of the Hessian has been widely observed in multiple circumstances (Sagun et al., 2016; Arjevani & Field, 2020; Pan et al., 2021). The power of this adaptation shines when adaptive gradient methods are employed.

We consider the standard stochastic approximation framework (Kushner & Clark, 2012) and denote the gradient noise at $\mathbf{w}$ to be $\mathbf{n}(\mathbf{w}; \xi) \triangleq \nabla f(\mathbf{w}) - \nabla_{\mathbf{w}} f(\mathbf{w}; \xi)$. We assume the following assumptions on gradient noise throughout this paper.

**Assumption 3.4** (Unbiased Independent gradient). Each $\nabla f(\mathbf{w}; \xi)$ is independently drawn and

$$\mathbb{E}[\mathbf{n}(\mathbf{w}; \xi)] = 0. \tag{4}$$

---

**Algorithm 1** Adagrad

---

1: **Input:** $\mathbf{w}_0 \in \mathbb{R}^d$, $\{\eta_t\}_{t=0}^{T-1} \in \mathbb{R}$, $\epsilon \in \mathbb{R}$, and batch size $M \in \mathbb{N}$
2: Initialize $\mathbf{v}_{-1} = \epsilon^2 \mathbf{1}_d$
3: **for** $t = 0$ **to** $T - 1$ **do**
4:      Sample mini-batch $\mathcal{B}_t$ with $|\mathcal{B}_t| \equiv M$ uniformly
5:      $\mathbf{g}_t = \frac{1}{M} \sum_{\xi \in \mathcal{B}_t} \nabla_\mathbf{w} f(\mathbf{w}_t; \xi)$
6:      $\mathbf{v}_t = \mathbf{v}_{t-1} + (\mathbf{g}_t \odot \mathbf{g}_t)$
7:      $\mathbf{\Lambda}_t = \text{diag}(\sqrt{\mathbf{v}_t})$
8:      **Option I:** $\mathbf{w}_{t+1} = \Pi_{\mathcal{W}}^{\mathbf{\Lambda}_t}(\mathbf{w}_t - \eta_t \mathbf{\Lambda}_t^{-1} \mathbf{g}_t)$
9:      **Option II:** $\mathbf{w}_{t+1} = \mathbf{w}_t - \eta_t \mathbf{\Lambda}_t^{-1} \mathbf{g}_t$
10: **end for**
11: **Output:** $1/T \sum_{t=0}^{T-1} \mathbf{w}_t$

---

**Assumption 3.5** (Anisotropic noise). *There exists positive vector* $\boldsymbol{\sigma} = [\boldsymbol{\sigma}_1, \ldots, \boldsymbol{\sigma}_d] \in \mathbb{R}_+^d$ *such that*

$$\mathbb{E}\left[\mathbf{n}_j(\mathbf{w}; \xi)^2\right] \leq \boldsymbol{\sigma}_j^2 \quad \text{for all} \quad j \in [d]. \tag{5}$$

Note that Assumption 3.5 implies the standard bounded noise assumption by $\mathbb{E}[\|\mathbf{n}(\mathbf{w}; \xi)\|_2^2] \leq \|\boldsymbol{\sigma}\|_2^2$. Intuitively, it upper bounds all the coordinates of $\mathbf{n}(\mathbf{w}; \xi)$ instead of only the norm and gives more detailed information on the scale of noise. Generally, Combining Assumption 3.3 and 3.5 allows more fine-grained analysis, which can take the sparsity of the model into account. Note that the anisotropic noise assumption has also been explored in other lines of studies on sign-based methods (Bernstein et al., 2018a;b; Crawshaw et al., 2022) and quadratics (Dieuleveut et al., 2017; Ge et al., 2019; Pan et al., 2021; 2023). It is also closely related to Assumption 2 in Zhang et al. (2020b), where the authors attempt to model the heavy-tailedness of neural networks.

## 4 ADAGRAD WITH ANISOTROPIC ASSUMPTIONS

How to accelerate the convergence of SGD has been a fundamental problem in training machine learning models. One possible approach is to lower the implicit dependence of dimension $d$. As discussed in Nguyen et al. (2019), in smooth strongly convex settings, the lower bound of SGD can be $d$ times larger than a wider class of algorithms including adaptive gradient methods. The original Adagrad paper (Duchi et al., 2011) showed that Adagrad has better dimension dependence than SGD when the stochastic gradients are generally sparse in the non-smooth convex scheme. However, in the stochastic smooth optimization scheme, to the best of our knowledge, existing theoretical results are insufficient to account for the benefit of adaptive gradient methods. We attempt to fill the gap, equipped with the anisotropic Assumptions 3.3 and 3.5.

### 4.1 CONVEX CASES

For a warmup, we present results in the convex case to first gain some intuition on how the anisotropic assumptions can better describe the convergence of AdaGrad in the large batch setting.

**Theorem 4.1** (Convex convergence of Adagrad). *Under Assumptions 3.1, 3.3, 3.4, 3.5, for the sequence $\{\mathbf{w}_t\}_{t=1}^T$ generated by Adagrad (Algorithm 1 with option I) with constant step size $\eta_t \equiv \eta = D_\infty$, it holds that for $\bar{\mathbf{w}}_T = (1/T) \sum_{t=0}^{T-1} \mathbf{w}_t$,*

$$\mathbb{E}[f(\bar{\mathbf{w}}_T) - f(\mathbf{w}_*)] = \mathcal{O}\left(\frac{D_\infty \|\boldsymbol{\sigma}\|_1}{\sqrt{MT}} + \frac{\|\mathbf{L}\|_1 D_\infty^2}{T}\right) + \mathcal{O}\left(\frac{\epsilon D_2^2}{D_\infty T}\right).$$

Note that $\epsilon$ is employed mainly for numerical stability and is commonly very small (in order $10^{-10}$ by default). Theorem 4.1 shows that the convergence of AdaGrad depends on $\|\mathbf{L}\|_1$ and $\|\boldsymbol{\sigma}\|_1$, which require Assumptions 3.3 and 3.5 to describe. In contrast, if we only use the standard $L$-smooth and $\sigma_2^2$-bounded gradient variance assumption, explicit dimensional dependence will be inevitably involved, resulting in worse results than coordinate-wise uniform step sizes like Vaswani et al. (2020) To better see how Theorem 4.3 can describe the potential benefits of AdaGrad, we also include the convergence

rates of SGD and AdaGrad-Norm as representatives of algorithms with coordinate-wisely uniform step sizes here[2]:

$$\text{SGD \& AdaGrad-Norm:} \quad \mathbb{E}\left[f(\mathbf{w}) - f(\mathbf{w}_*)\right] = \mathcal{O}\left(\frac{D_2 \left\|\boldsymbol{\sigma}\right\|_2}{\sqrt{MT}} + \frac{\left\|\mathbf{L}\right\|_\infty D_2^2}{T}\right). \tag{6}$$

By comparing the results in Theorem 4.3 and (6), we can find that whether AdaGrad is better than SGD or AdaGrad-Norm largely relies on the ratios $\frac{D_\infty \left\|\boldsymbol{\sigma}\right\|_1}{D_2 \left\|\boldsymbol{\sigma}\right\|_2}$ and $\frac{\left\|\mathbf{L}\right\|_1 D_\infty^2}{\left\|\mathbf{L}\right\|_\infty D_2^2}$, which reflect the sparsity of the curvature, noise, and the geometry of $\mathcal{W}$. When $M$ is small such that the variance term, i.e. the term relevant with $\boldsymbol{\sigma}$, is dominant, our conclusion is consistent with that presented in Duchi et al. (2011) for nonsmooth cases. Generally, when (1) $\boldsymbol{\sigma}$ is sparse, i.e. has very different scales in different coordinates, which implies that $\left\|\boldsymbol{\sigma}\right\|_1 \ll \sqrt{d} \left\|\boldsymbol{\sigma}\right\|_2$; (2) $\mathcal{W}$ satisfies that $D_2$ is close to $\sqrt{d} D_\infty$, which can be satisfied by setting $\mathcal{W}$ to be a hypercube, the variance term of Adagrad might be much smaller than that of SGD. When a large batch size is employed, the bias term, the $\mathcal{O}(1/T)$ term irrelevant with noise, can also be important, and thus the superior performance of Adagrad in this case additionally requires that (3) $\mathbf{L}$ is sparse such that $\left\|\mathbf{L}\right\|_1 \ll d \left\|\mathbf{L}\right\|_\infty$. Following Duchi et al. (2011), we also provide a concrete example for better understanding the quantities in Appendix B.

*Remark* 4.2. It is also worth pointing out that the ratio between bias terms of Theorem 4.1 and (6) can be in order $\Theta(1/d)$ in extreme cases, while the variance ratio can only be $\Theta(1/\sqrt{d})$. This suggests an even sharper possible gap when $M$ is large and might provide some intuition on the observation that adaptive gradient methods benefit more from large batch size than SGD (Kunstner et al., 2023).

### 4.2 NONCONVEX CASES

Next we consider the more general nonconvex scheme.

**Theorem 4.3** (Nonconvex convergence of Adagrad). *Under Assumptions 3.2, 3.3, 3.4, 3.5, for the sequence $\{\mathbf{w}_t\}_{t=1}^{T-1}$ generated by Adagrad (Algorithm 1 with option II) with constant step size $\eta_t \equiv \eta = \sqrt{\frac{\left\|\mathbf{L}\right\|_1}{\Delta}}$, it holds that*

$$\frac{1}{T}\left(\mathbb{E}\left[\sqrt{\sum_{t=0}^{t-1} \left\|\nabla f(\mathbf{w}_t)\right\|_1^2}\right]\right)^2 = \tilde{\mathcal{O}}\left(\frac{\sqrt{\left\|\mathbf{L}\right\|_1 \Delta} \left\|\boldsymbol{\sigma}\right\|_1}{\sqrt{MT}} + \frac{\left\|\boldsymbol{\sigma}\right\|_1^2}{M\sqrt{T}} + \frac{\left\|\mathbf{L}\right\|_1 \Delta}{T}\right)$$
$$+ \tilde{\mathcal{O}}\left(\frac{d\epsilon\sqrt{\left\|\mathbf{L}\right\|_1 \Delta}}{T} + \frac{d\epsilon \left\|\boldsymbol{\sigma}\right\|_1}{\sqrt{MT}}\right),$$

*where $\Delta = f(\mathbf{w}_0) - f^*$ and we use $\tilde{\mathcal{O}}(\cdot)$ to hide logarithmic factors.*

It is worth pointing out that Theorem 4.3 obtains convergence of $\left\|\nabla f(\mathbf{w})\right\|_1$ instead of the common $\left\|\nabla f(\mathbf{w})\right\|_2$ by algorithms with coordinate-wise uniform step sizes like SGD, which indicates at most $\sqrt{d}$ times tighter results when the gradients are generally dense. Similar to the convex case, the introduction of the anisotropic assumptions 3.3 and 3.5 removes the explicit dependence on dimension $d$ compared to existing results on adaptive gradient methods like Défossez et al. (2020); Liu et al. (2023b); Zhang et al. (2022); Wang et al. (2022), rendering AdaGrad at least comparable to algorithms with coordinate-wisely uniform step sizes like SGD, which has the following results:

$$\text{SGD:} \quad \mathbb{E}\left[\left\|\nabla f(\mathbf{w})\right\|_2^2\right] \leq \mathcal{O}\left(\frac{\sqrt{\left\|\mathbf{L}\right\|_\infty \Delta} \left\|\boldsymbol{\sigma}\right\|_2}{\sqrt{MT}} + \frac{\left\|\mathbf{L}\right\|_\infty \Delta}{T}\right) \tag{7}$$

We can find that when the batch size is large enough such that $M \geq \left\|\boldsymbol{\sigma}\right\|_1^2 / (\left\|\mathbf{L}\right\|_1 \Delta)$, the comparison between Theorem 4.3 and (7) is generally consistent with the comparison between SGD and SignSGD (Bernstein et al., 2018a). It mainly relies on the sparsity of $\nabla f(\mathbf{w}_t)$, $\mathbf{L}$, and $\boldsymbol{\sigma}$, which can determine the ratio between the upper bounds. Generally speaking, when $\mathbf{L}$ and $\boldsymbol{\sigma}$ are sparse and the gradients $\nabla f(\mathbf{w}_t)$ are relatively dense, Adagrad shines compared to SGD in terms of having

---

[2]We also include the standard proof of SGD and discussions on convex SGD convergence in Appendix D for completeness. The result of AdaGrad-Norm can be found in Levy et al. (2018).

Table 1: We summarize the convergence rates of algorithms under the nonconvex smooth settings, with large enough batch size such that $M \geq \|\boldsymbol{\sigma}\|_1^2 / (\|\mathbf{L}\|_1 \Delta)$. Note that we omit logarithmic terms here. The convergence of SGD in the smooth setting is a standard result, and we include the theorem in the appendix for reference. Also, we leave the convergence of SGD under generalized smoothness blank as it is generally incomparable with other results, which we will discuss in Section 5.

| Algorithms | Convergence Objective | Anisotropic Smooth | Anisotropic Generalized Smooth |
|---|---|---|---|
| SGD | $\mathbb{E}\left[\|\nabla f(\mathbf{w})\|_2^2\right]$ | $\frac{\sqrt{\|\mathbf{L}\|_\infty \Delta}\|\boldsymbol{\sigma}\|_2}{\sqrt{MT}} + \frac{\|\mathbf{L}\|_\infty \Delta}{T}$ 
 Theorem D.3 | – |
| AdaGrad-Norm | $\left(\mathbb{E}\left[\|\nabla f(\mathbf{w})\|_2\right]\right)^2$ | $\frac{\sqrt{\|\mathbf{L}\|_\infty \Delta}\|\boldsymbol{\sigma}\|_2}{\sqrt{MT}} + \frac{\|\mathbf{L}\|_\infty \Delta}{T}$ 
 (Faw et al., 2023) | $\frac{\sqrt{L_0 \Delta}\|\boldsymbol{\sigma}\|_2}{\sqrt{MT}} + \frac{L_0 \Delta}{T}$ 
 $+\frac{L_1 \Delta\|\boldsymbol{\sigma}\|_2}{\sqrt{MT}} + \frac{L_1^2 \Delta^2}{T}$ 
 (Faw et al., 2023) |
| AdaGrad | $\left(\mathbb{E}\left[\|\nabla f(\mathbf{w})\|_1\right]\right)^2$ | $\frac{\sqrt{\|\mathbf{L}\|_1 \Delta}\|\boldsymbol{\sigma}\|_1}{\sqrt{MT}} + \frac{\|\mathbf{L}\|_1 \Delta}{T}$ 
 Theorem 4.3 | $\frac{\sqrt{\|\mathbf{L}_0\|_1 \Delta}\|\boldsymbol{\sigma}\|_1}{\sqrt{MT}} + \frac{\|\mathbf{L}_0\|_1 \Delta}{T}$ 
 $+\frac{\|\mathbf{L}_1\|_\infty \Delta\|\boldsymbol{\sigma}\|_1}{\sqrt{MT}} + \frac{\|\mathbf{L}_1\|_\infty^2 \Delta^2}{T}$ 
 Theorem 5.3 |

a tighter convergence guarantee. We note that this can be a realistic case, as $\mathbf{L}$ can be extremely sparse based on many observations on multiple scenarios (Sagun et al., 2016; Arjevani & Field, 2020; Pan et al., 2021), and on the other side, $\frac{\|\boldsymbol{\sigma}\|_1}{\|\boldsymbol{\sigma}\|_2}$ and $\frac{\|\nabla f(\mathbf{w}_t)\|_1}{\|\nabla f(\mathbf{w}_t)\|_2}$ can be mild constants as examined by Bernstein et al. (2018a). Therefore, with all these conditions satisfied, we show the potentially faster convergence of AdaGrad compared to SGD. Also, the consistency between the convergence results of AdaGrad and SignSGD provides theoretical insights into the close relation between adaptive gradient methods and sign-based methods.

## 5 ADAGRAD WITH GENERALIZED ANISOTROPIC SMOOTHNESS

In previous sections, we have discussed how AdaGrad can potentially outperform algorithms with uniform step sizes under the anisotropic smoothness settings. However, the loss function in practice can commonly dissatisfy the smoothness assumption, with local smoothness potentially unbounded. To better describe the complicated real cases, Zhang et al. (2019) introduce the $(L_0, L_1)$-smoothness as a generalization of the standard $L$-smoothness, which can be written as

$$\|\nabla f(\mathbf{w}) - \nabla f(\mathbf{w}')\|_2 \leq (L_0 + L_1 \|\nabla f(\mathbf{w})\|_2) \|\mathbf{w} - \mathbf{w}'\|_2 \qquad (8)$$

for all $\|\mathbf{w} - \mathbf{w}'\| \leq 1/L_1$ (Zhang et al., 2020a). By involving the gradient term to describe the local smoothness, this assumption can even be applicable to describe neural networks (Zhang et al., 2019; Crawshaw et al., 2022). In this section, we discuss an extension of both the anisotropic $\mathbf{L}$-smoothness and the $(L_0, L_1)$-smoothness and the corresponding convergence results of AdaGrad.

**Assumption 5.1** (Anisotropic $(\mathbf{L}_0, \mathbf{L}_1)$-smoothness). There exists positive vectors $\mathbf{L}_0 = [\mathbf{L}_{0,1}, \ldots, \mathbf{L}_{0,d}] \in \mathbb{R}_+^d$ and $\mathbf{L}_1 = [\mathbf{L}_{1,1}, \ldots, \mathbf{L}_{1,d}] \in \mathbb{R}_+^d$ such that $f(\cdot)$ is $(\mathbf{L}_0, \mathbf{L}_1)$-smooth, namely, for $\mathbf{w}, \mathbf{w}' \in \mathcal{W}$ such that $\|\mathbf{w} - \mathbf{w}'\|_{\mathbf{L}_1} \leq \sqrt{d}$, it holds that

$$\|\nabla f(\mathbf{w}) - \nabla f(\mathbf{w}')\|_{(\mathbf{L}(\mathbf{w}))^{-1}} \leq \|\mathbf{w} - \mathbf{w}'\|_{\mathbf{L}(\mathbf{w})}, \qquad (9)$$

where $\mathbf{L}(\mathbf{w}) = [[\mathbf{L}(\mathbf{w})]_1, \ldots, [\mathbf{L}(\mathbf{w})]_d] \in \mathbb{R}^d$ and $[\mathbf{L}(\mathbf{w})]_j = \mathbf{L}_{0,j} + \mathbf{L}_{1,j} |\partial_j f(\mathbf{w})|$ for all $j \in [d]$.

For one thing, Assumption 5.1 is a natural generalization of Assumption 3.3 and can directly imply it by setting $\mathbf{L}_1 = 0$, hence also enjoys the nice properties we have discussed for Assumption 3.3. Also, in the spirits of (8), we include the gradient magnitudes to describe the local smoothness. Instead of the gradient 2-norm, we use the absolute value of each coordinate of the gradient, which is intuitively more relevant to the anisotropic nature of curvature. We also conducted numerical experiments to verify the validity of the assumption, as shown in Figure 1.

*Remark* 5.2. It is also worth noticing that a similar but different coordinate-wise generalized smoothness assumption has been considered in Crawshaw et al. (2022):

$$|\partial_j f(\mathbf{w}) - \partial_j f(\mathbf{w}')| \leq (\mathbf{L}_{0,j} + \mathbf{L}_{1,j} |\partial_j f(\mathbf{w})|) \|\mathbf{w} - \mathbf{w}'\|_2 \tag{10}$$

for all $\|\mathbf{w} - \mathbf{w}'\|_2 \leq 1/ \|\mathbf{L}_1\|_\infty$ and all coordinates $j \in [d]$. Compared to this assumption, Assumption 5.1 offers several evident advantages: (1) Assumption 5.1 does not require conditions for each coordinate; (2) Experiments show that Assumption 5.1 can well describe the real anisotropic curvature; (3) Technically, it is difficult to avoid explicit existence of $d$ in the final convergence bound of AdaGrad if Eqn. (10) is employed, for instance, even the convergence of the Generalized SignSGD obtained in Crawshaw et al. (2022) has explicit dependence on $d$.

**Theorem 5.3** (Convergence of Adagrad with generalized smoothness). *Under Assumptions 3.2, 3.4, 3.5, 5.1, for the sequence $\{\mathbf{w}_t\}_{t=1}^{T-1}$ generated by Adagrad (Algorithm 1 with option II) with constant step size $\eta_t \equiv \eta = \min \left\{ \frac{1}{4\|\mathbf{L}_1\|_\infty}, \sqrt{\frac{\|\mathbf{L}_0\|_1}{\Delta}} \right\}$, it holds that*

$$\frac{1}{T} \left( \mathbb{E} \left[ \sqrt{\sum_{t=0}^{t-1} \|\nabla f(\mathbf{w}_t)\|_1^2} \right] \right)^2 = \tilde{\mathcal{O}} \left( \frac{\sqrt{\|\mathbf{L}_0\|_1 \Delta} \|\boldsymbol{\sigma}\|_1}{\sqrt{MT}} + \frac{\|\boldsymbol{\sigma}\|_1^2}{M\sqrt{T}} + \frac{\|\mathbf{L}_0\|_1 \Delta}{T} \right)$$

$$+ \tilde{\mathcal{O}} \left( \frac{\|\mathbf{L}_1\|_\infty \Delta \|\boldsymbol{\sigma}\|_1}{\sqrt{MT}} + \frac{\|\mathbf{L}_1\|_\infty^2 \Delta^2}{T} \right)$$

$$+ \tilde{\mathcal{O}} \left( \frac{d\epsilon \left( \sqrt{\|\mathbf{L}_0\|_1 \Delta} + \|\mathbf{L}_1\|_\infty \Delta \right)}{T} + \frac{d\epsilon \|\boldsymbol{\sigma}\|_1}{\sqrt{MT}} \right),$$

*where $\Delta = f(\mathbf{w}_0) - f^*$ and we use $\tilde{\mathcal{O}}(\cdot)$ to hide logarithmic factors.*

Note that Theorem 5.3 is a generalization of Theorem 4.3 based on the relation between Assumptions 5.1 and 3.3, which can directly imply Theorem 4.3 by simply setting $\mathbf{L}_1 = 0$. Intuitively, the interpretation of the theorem leads to the following implications.

**Comparison with SGD.** Zhang et al. (2019) show that SGD generally requires an additional assumption on universally bounded gradients to obtain convergence under the generalized $(L_0, L_1)$-smoothness, and thus clipping should be introduced. The theoretical properties of clipped SGD has been extensively studied after that. However, to the best of our knowledge, existing results on clipped SGD either require restrictive noise assumptions (Zhang et al., 2019; 2020a; Chen et al., 2020; Qian et al., 2021) or can only shows worse rate on $T$ and $M$ (Koloskova et al., 2023; Gorbunov et al., 2020). These results are generally incomparable with Theorem 4.3, showing the superiority of adaptive gradient methods, as also found in existing results (Wang et al., 2023; Li et al., 2024).

**Comparison with AdaGrad-Norm.** One can also look at the convergence results of AdaGrad-Norm (Streeter & McMahan, 2010), a scalar step size variant of AdaGrad, with assumption (8):

$$\text{AdaGrad-Norm:} \quad (\mathbb{E} [\|\nabla f(\mathbf{w})\|_2])^2 \leq \tilde{\mathcal{O}} \left( \frac{\sqrt{L_0\Delta} \|\boldsymbol{\sigma}\|_2}{\sqrt{MT}} + \frac{L_0\Delta}{T} + \frac{L_1\Delta \|\boldsymbol{\sigma}\|_2}{\sqrt{MT}} + \frac{L_1^2\Delta^2}{T} \right),$$

which is obtained by taking $\eta$ to optimize the upper bound in Theorem 3 in Faw et al. (2023). It is worth noticing that in the case $\mathbf{L}_1 = 0$, the comparison between AdaGrad and AdaGrad-Norm is generally consistent with that between AdaGrad and SGD discussed in Section 4, showing AdaGrad's effectiveness for anisotropic problems. On the other hand, when it comes to the general $(\mathbf{L}_0, \mathbf{L}_1)$-smoothness assumption, things become more complicated given the indeterminate relationship between the anisotropic $\mathbf{L}_1$ and $L_1$ in (8). This relation is mainly about whether Assumption 5.1 or the initial $(L_0, L_1)$-smoothness assumption can better fit the common training settings, such as large-scale language model pre-training, which is an intriguing topic worth exploring in the future.

*Remark* 5.4. *(Technical Contribution)* From the technical perspective, our proof generally considers a similar main line to Défossez et al. (2020); Ward et al. (2020) but involves multiple novel proof techniques to remove the restrictive assumption on globally bounded $\|\nabla f(\mathbf{w})\|$ and enable the presence of generalized smoothness. Also, Theorem 4.3 can directly imply a comparable convergence result of AdaGrad-Norm with Faw et al. (2023); Wang et al. (2023) while it avoids the complicated proof in Faw et al. (2023) and the heavy dependence on $1/\epsilon$ in Wang et al. (2023). On the other hand, however, Faw et al. (2023); Wang et al. (2023) allows relaxed noise assumption, which may be of interest and we leave it for possible future work.

## 6 EXPERIMENTAL RESULTS

### 6.1 CONVEX CASE

To verify the aforementioned theoretical results, we conduct experiments on logistics regressions, whose loss functions are generally convex smooth. Specifically, we utilize real-world datasets `a4a`, `a6a`, `a9a`, `real-sim` and `rcv1.binary` from `libsvm` (Chang & Lin, 2011), which comprises of $N = 4781, 11220, 32561, 20242$ and $72309$ samples respectively. Within the former three datasets, each sample has a feature of $d = 123$ dimensions, which are generally sparse given only 14 non-zero-valued dimensions in average for each sample. For the latter two large datasets, each sample possesses $d = 47,236$ and $d = 20,958$ feature dimensions individually in each dataset, where only $51.29$ and $74.05$ dimensions in average are non-zero. More details of the experimental setup are available in Appendix A.

Table 2: Statistics on logistic regression. $D_2 = \max_t \|\mathbf{w}_t - \mathbf{w}_*\|_2$ and $D_\infty = \max_t \|\mathbf{w}_t - \mathbf{w}_*\|_\infty$ are estimated by the maximum value under all searched settings without loss explosion. Smaller values in $C_{\mathbf{var}} \triangleq D_\infty/D_2$ represent better theoretical bounds for Adagrad when compared with SGD. It is evident that in large sparse datasets, $D_\infty$ is much smaller than $D_2$, verifying the empirical gains of Adagrad implied by our theory.

| DATASET | $D_\infty$ | $D_2$ | $\|\mathbf{L}\|_1$ | $\|\mathbf{L}\|_\infty$ | $C_{\mathbf{VAR}}$ |
|---------|-----------|-------|-----------|----------------|----------|
| A4A | 9.73 | 32.24 | 14.87 | 7.26 | **0.30** |
| A6A | 8.88 | 28.57 | 14.87 | 7.26 | **0.31** |
| A9A | 9.79 | 29.43 | 14.87 | 7.27 | **0.32** |
| REAL-SIM | 34.70 | 729.50 | 2.00 | 1.01 | **0.05** |
| RCV1.BIN | 15.32 | 635.47 | 2.00 | 1.02 | **0.02** |

As shown by the results presented in Table 3, for cases when the variance term dominates, e.g. `a9a`, Adagrad demonstrates similar convergence behaviors for varied batch sizes. This behavior cannot be explained by previous nonsmooth theories, since $T = N/M$ should provide worse convergence guarantees $\mathcal{O}(1/\sqrt{T})$ when batch size $M$ increases. Furthermore, Adagrad constantly provides faster convergence than SGD, which verifies the superiority suggested in our theorems. In addition, when the batch size increases and the bias term becomes dominant, Adagrad is affected less than SGD, showing its robustness against different batch sizes.

Table 3: Training losses of SGD and Adagrad on logistic regression over 3 seeds with batch size $M$. Note that Adagrad's convergence behavior is generally unaffected by the batch size when $M \leq \sqrt{N}$.

| DATASET (SMALL) | METHOD | $(f(w) - f(w_*)) \times 10^{-3}$ | | | | | |
|---|---|---|---|---|---|---|---|
| | | $M = 1$ | $M = 4$ | $M = 16$ | $M = 64$ | $M = 256$ | $M = 1024$ |
| A4A | SGD | 2.66±0.08 | 2.32±0.34 | 3.47±0.07 | 2.24±0.14 | 4.61±0.03 | 8.47±0.05 |
| | ADAGRAD | 0.22±0.03 | 0.22±0.03 | 0.25±0.03 | 0.24±0.03 | 0.29±0.03 | 0.51±0.08 |
| A6A | SGD | 0.87±0.09 | 1.40±0.01 | 0.82±0.09 | 1.56±0.03 | 1.03±0.01 | 2.53±0.03 |
| | ADAGRAD | 0.16±0.00 | 0.16±0.00 | 0.16±0.01 | 0.21±0.05 | 0.17±0.01 | 0.20±0.02 |
| A9A | SGD | 0.77±0.08 | 0.47±0.01 | 0.48±0.05 | 0.58±0.01 | 0.52±0.06 | 0.76±0.01 |
| | ADAGRAD | 0.14±0.01 | 0.14±0.00 | 0.15±0.00 | 0.16±0.01 | 0.20±0.01 | 0.12±0.02 |
| DATASET (LARGE) | METHOD | $(f(w) - f(w_*)) \times 10^{-1}$ | | | | | |
| | | $M = 1$ | $M = 4$ | $M = 16$ | $M = 64$ | $M = 256$ | $M = 1024$ |
| REAL-SIM | SGD | 0.42±0.08 | 0.27±0.08 | 0.52±0.06 | 0.92±0.03 | 1.57±0.02 | 2.68±0.01 |
| | ADAGRAD | 0.14±0.00 | 0.14±0.00 | 0.14±0.00 | 0.14±0.00 | 0.15±0.00 | 0.19±0.00 |
| RCV1.BINARY | SGD | 0.48±0.02 | 0.28±0.07 | 0.68±0.04 | 1.33±0.04 | 2.55±0.21 | 5.02±0.16 |
| | ADAGRAD | 0.10±0.00 | 0.10±0.00 | 0.10±0.00 | 0.11±0.00 | 0.14±0.01 | 0.20±0.01 |

## 6.2 NONCONVEX CASE

For nonconvex cases, we check the instruction-following fine-tuning task on Alpaca (Taori et al., 2023) dataset with GPT-2 (Radford et al., 2019) model. GPT-2 utilizes GELU (Hendrycks & Gimpel, 2016) as its activation function. As shown in Table 4, it can be observed that Adagrad still outperforms SGD with different batch sizes. The loss gap is especially salient under large batch sizes. This also confirms that Adagrad's convergence speed is not significantly affected by the large batch size, which matches the results in our theory. Full experimental details are available in Appendix A.

Table 4: Losses on instruction following tasks with dataset Alpaca and GPT2 model.

| METHOD | TRAINING LOSS $f(w)$ | | |
|---|---|---|---|
| | $M = 32$ | 64 | 128 |
| SGD | 2.20 | 2.24 | 2.29 |
| ADAGRAD | 2.14 | 2.12 | 2.11 |
| | $M = 256$ | 512 | 1024 |
| SGD | 2.36 | 2.45 | 2.57 |
| ADAGRAD | 2.12 | 2.14 | 2.20 |

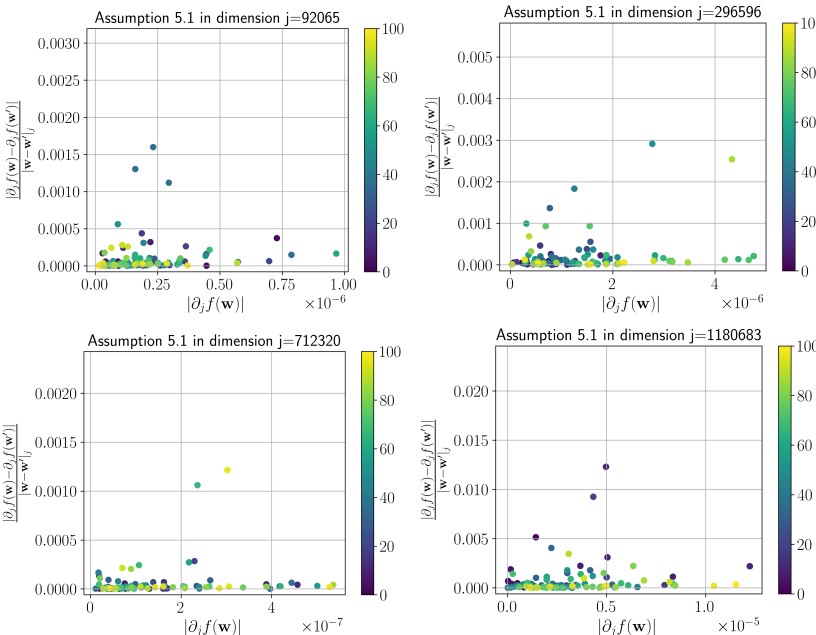

Figure 1: Verification of Assumption 5.1 in GPT-2 on Alpaca dataset. **x-axis**: $|\partial_j f(\mathbf{w})|$, **y-axis**: $|\partial_j f(\mathbf{w}) - \partial_j f(\mathbf{w}')|/|\mathbf{w} - \mathbf{w}'|_j$, where the color represents the iteration index of $\mathbf{w}$. We run Adam with full gradients for 100 steps and randomly selected nearby points $\mathbf{w}$ and $\mathbf{w}$' along the trajectory to plot the scatter points.

We also examine Assumption 5.1 under this non-convex setting by following the common practice of (Zhang et al., 2019; Crawshaw et al., 2022). Due to the inherent dependency on dimensions in Assumption 5.1, we randomly sample a small portion of the parameters in GPT-2 to check. As shown in Figure 1, a linear bound of local smoothness with respect to the gradient value holds in all randomly chosen dimensions. Notice that the observed condition in Figure 1 is even stronger, as it directly implies Assumption 5.1 by summing up all the dimensions.

## 7 CONCLUSION

In this paper, we present the theoretical convergence results of Adagrad under anisotropic smoothness and noise assumptions. We further introduce a novel anisotropic generalized smoothness assumption to extend the aforementioned results in a more fine-grained analysis. Based on the theorems, we conduct comparisons between the convergence rates of AdaGrad and SGD and AdaGrad-Norm in the large batch settings, which provides a deeper theoretical understanding of when and why adaptive gradient methods can outperform classical gradient algorithms with coordinate-wise uniform step sizes. Empirical studies offer strong evidence to support our proposed assumptions and theory.

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

# A MORE EXPERIMENTAL DETAILS

## A.1 CONVEX CASE

For `a4a`, `a6a` and `a9a`, we run 100 epochs of optimization with SGD and Adagrad individually, starting from a uniform distribution initialization $\mathbf{w}_0 \in \mathcal{U}(-0.05, 0.05)^d$. To find the best hyperparameter, grid searches are conducted for both algorithms, with the search space being initial learning rate $\eta \in \{10.0, 1.0, 0.1, 0.01\}$ and learning rate schedules being either constant $\eta_t \equiv \eta$ or inverse square root decay $\eta_t = \eta/\sqrt{t+1}$, the same choices as in our theorems. For large datasets `real-sim` and `rcv1.binary`, all settings stay the same, except for the number of epochs 3 and the initialization $\mathbf{w}_0 \in \mathcal{U}(-1/d, 1/d)^d \times 10^{-2}$.

Since it is generally hard to obtain analytical closed-form solutions for logistic regressions, we run gradient descent for $10^6$ epochs to obtain an approximated optimum $w_*$ for small datasets `a4a`, `a6a` and `a9a`. For larger ones `real-sim` and `rcv1.binary`, we run $10^2$ epochs of Adagrad instead, since GD converges much slower in comparison. In addition, for computing $\|\mathbf{H}\|_2$ in large datasets, we run 10 iterations of power iteration to approximate the largest eigenvalue, which quickly converge to desired precisions $\leq 10^{-5}$.

Table 5: Statistics on logistic regression. $D_2 = \max_t \|\mathbf{w}_t - \mathbf{w}_*\|_2$ and $D_\infty = \max_t \|\mathbf{w}_t - \mathbf{w}_*\|_\infty$ are estimated by the maximum value under all searched settings without loss explosion. Smaller values in $C_{\mathbf{var}} \triangleq D_\infty/D_2$ represent better theoretical bounds for Adagrad when compared with SGD, and $C_{\mathbf{bias}} \triangleq (\|\mathbf{L}\|_1 D_\infty^2)/(\|\mathbf{L}\|_\infty D_2^2)$ corresponds to the bias difference.

| DATASET | $D_\infty$ | $D_2$ | $\|\mathbf{L}\|_1$ | $\|\mathbf{L}\|_\infty$ | $C_{\text{VAR}}$ | $C_{\text{BIAS}}$ | $\min_t f(\mathbf{w}_t) - f(\mathbf{w}_*)$ | |
|---|---|---|---|---|---|---|---|---|
| | | | | | | | SGD | ADAGRAD |
| A4A | 9.73 | 32.24 | 14.87 | 7.26 | 0.30 | 0.19 | $1.46 \times 10^{-3}$ | $6.88 \times 10^{-5}$ |
| A6A | 8.88 | 28.57 | 14.87 | 7.26 | 0.31 | 0.20 | $4.36 \times 10^{-4}$ | $6.39 \times 10^{-5}$ |
| A9A | 9.79 | 29.43 | 14.87 | 7.27 | 0.32 | 0.23 | $3.47 \times 10^{-4}$ | $5.04 \times 10^{-5}$ |

## A.2 NONCONVEX CASE

For all experiments, we run 3 epochs of optimization with SGD and Adagrad, which is one of the recommended settings of the Alpaca dataset (Taori et al., 2023) (`https://github.com/tatsu-lab/stanford_alpaca?tab=readme-ov-file#fine-tuning`). We search the learning rate $\eta \in \{1.0, 10^{-1}, 10^{-2}, 10^{-3}, 10^{-4}, 10^{-5}, 10^{-6}\}$ and report the best result in training loss for both SGD and Adagrad. The maximum sequence length is set to 512, along with the learning rate schedule being set to cosine decay (Loshchilov & Hutter, 2016). Other settings remain the same as default ones in huggingface `transformers` (`https://github.com/huggingface/transformers`). In all our implementations, we use the version `transformers==4.38.2`. All experiments are conducted on a single A40 GPU, where gradient accumulation is adopted for batch sizes larger than 128 to reduce memory cost.

In addition to the presented results, we include in Figure 2 the loss curves of SGD and Adagrad under batch sizes 256 and 512. They represent the typical loss decrease tendency of all experiments, where Adagrad converges faster than SGD since the beginning, and converges to a point with smaller loss.

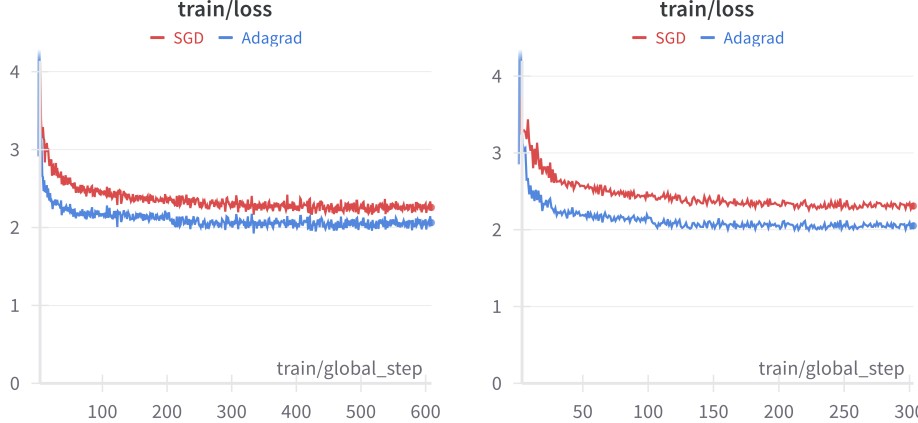

Figure 2: Training loss curves of SGD and Adagrad for instruction following tasks on Alpaca with GPT2. **Left:** batch size 256, **Right:** batch size 512.

Regarding licenses, the Alpaca dataset is released under Creative Commons Attribution-NonCommercial 4.0 International Public License (`https://github.com/tatsu-lab/stanford_alpaca/blob/main/DATA_LICENSE`), while GPT-2 is released under MIT License (`https://huggingface.co/openai-community/gpt2`). The code repository of huggingface 'transformers' is released under Apache License 2.0 (`https://github.com/huggingface/transformers/blob/main/LICENSE`).

## B  AN EXAMPLE IN THE CONVEX CASE

Following the SVM example in Duchi et al. (2011), we provide Example B.1 to create a concrete case to help our illustration.

**Example B.1.** Consider a finite-sum optimization problem in a hypercube $\mathcal{W}$:

$$f(\mathbf{w}) = \frac{1}{n} \sum_{i=1}^{n} \phi_i(\mathbf{x}_i^\top \mathbf{w}),$$

where $\phi_i : \mathbb{R} \to \mathbb{R}$ are convex and twice-differentiable functions and $\mathbf{x}_i \in \mathbb{R}^d, \mathbf{w} \in \mathcal{W}$. We assume that the first and second order derivatives are bounded such that $|\phi_i'(\mathbf{w})| \leq G_1$ and $\phi_i''(\mathbf{w}) \leq G_2$ for all $i$ and $\mathbf{w} \in \mathcal{W}$. We also assume that the data points $\{\mathbf{x}_i\}_{i=1}^{n}$ yield that

$$\left[ \sum_{j=1}^{d} \left[ \frac{1}{n} \sum_{i=1}^{n} \mathbf{x}_{i,j}^2 \right]^{\frac{p}{2}} \right]^{\frac{1}{p}} \leq M_p,$$

for $p \geq 1$ and denote $\mathbf{H} = 1/n \sum_{i=1}^{n} \mathbf{x}_i \mathbf{x}_i^\top$. In this case, stochastic gradient yields that

$$\nabla_{\mathbf{w}} f(\mathbf{w}; \xi) = \mathbf{x}_\xi \cdot \phi_\xi'(\mathbf{x}_\xi^\top \mathbf{w}),$$

where $\xi$ is uniformly sampled from $\{1, \cdots, n\}$.

Note that Example B.1 includes a wide range of problems such as linear and logistic regression. We can check that the problem in Example B.1 yields Assumptions of Theorem D.1 and 4.1. As the stochastic gradients are sampled uniformly with replacement, the variance of gradient noise is reduced by factor $M$, where $M$ is the batch size, as proven in Lemma C.2. We also assume an exponential tail to quantify the imbalance of Hessian in our following example.

**Assumption B.2.** We say the problem has an exponential tail when there exists some constants $\tau > 1/d$ such that

$$\mathbf{H}_{j,j} = \frac{1}{n}\sum_{i=1}^{n}\mathbf{x}_{i,j}^2 \propto \exp\left(-\tau j\right)$$

for all $j = 1, \cdots, d$, where without loss of generality, we assume that $\mathbf{H}_{j,j} \geq \mathbf{H}_{k,k}$ for all $j \leq k$.

Note that the exponential tail is a typical example of highly imbalanced distributions of data, which is common in natural language modeling Zipf (2016), social networks Muchnik et al. (2013), and recommender systems Yin et al. (2012). Then recall the comparison between AdaGrad and SGD in the convex case:

$$\text{AdaGrad:} \quad \mathbb{E}\left[f(\bar{\mathbf{w}}_T) - f(\mathbf{w}_*)\right] = \mathcal{O}\left(\frac{D_\infty \left\|\boldsymbol{\sigma}\right\|_1}{\sqrt{MT}} + \frac{\left\|\mathbf{L}\right\|_1 D_\infty^2}{T}\right)$$

$$\text{SGD \& AdaGrad-Norm:} \quad \mathbb{E}\left[f(\bar{\mathbf{w}}_T) - f(\mathbf{w}_*)\right] = \mathcal{O}\left(\frac{D_2 \left\|\boldsymbol{\sigma}\right\|_2}{\sqrt{MT}} + \frac{\left\|\mathbf{L}\right\|_\infty D_2^2}{T}\right)$$

and we take the following ratios to denote the difference in bias and variance terms separately.

$$R_1 \triangleq \frac{D_\infty \left\|\boldsymbol{\sigma}\right\|_1}{D_2 \left\|\boldsymbol{\sigma}\right\|_2}, \quad R_2 \triangleq \frac{\left\|\mathbf{L}\right\|_1 D_\infty^2}{\left\|\mathbf{L}\right\|_\infty D_2^2}.$$

As we take $\mathcal{W}$ to be a hypercube, it holds that $D_2^2 = dD_\infty^2$. Concerning $R_1$, we have

$$\left\|\boldsymbol{\sigma}\right\|_1 \leq \max_{\mathbf{w}\in\mathcal{W}}\sum_{j=1}^{d}\sqrt{\frac{1}{n}\sum_{i=1}^{n}\left[\phi_i'(\mathbf{x}_i^\top\mathbf{w})\right]^2\mathbf{x}_{i,j}^2} \leq G_1 M_1$$

and similarly,

$$\left\|\boldsymbol{\sigma}\right\|_2 \leq \max_{\mathbf{w}\in\mathcal{W}}\sqrt{\frac{1}{n}\sum_{i=1}^{n}\sum_{j=1}^{d}\left[\phi_i'(\mathbf{x}_i^\top\mathbf{w})\right]^2\mathbf{x}_{i,j}^2} \leq G_1 M_2.$$

Therefore, based on the heavy-tailed assumption (Assumption B.2), we can obtain that

$$R_1 = \sqrt{\frac{\phi(\boldsymbol{\sigma})}{\phi(D)}} = \frac{M_1 D_\infty}{M_2 D_2} = \frac{M_1}{\sqrt{d}M_2} \propto \frac{\sum_{j=1}^{d}\exp\left(-\frac{1}{2}\tau j\right)}{\sqrt{d\sum_{j=1}^{d}\exp\left(-\tau j\right)}} \leq \frac{\sqrt{1-\exp\left(-\tau\right)}}{1-\exp\left(-\frac{1}{2}\tau\right)}\cdot\frac{1}{\sqrt{d}}.$$

If we consider $\tau$ to be a mild constant, it should be clear that $R_1$ can be close to $1/\sqrt{d}$ and $R_1 \ll 1$. On the other side, we have for all $\mathbf{w} \in \mathcal{W}$,

$$\mathbf{0} \preceq \nabla^2 f(\mathbf{w}) = \frac{1}{n}\sum_{i=1}^{n}\phi_i''(\mathbf{w})\cdot\mathbf{x}_i\mathbf{x}_i^\top \preceq G_2\mathbf{H}.$$

Moreover, that the diagonal matrix $\mathbf{L}_m = \text{diagonal}(\mathbf{L})$ satisfies $\mathbf{L}_m \succeq G_2\mathbf{H}$ is a sufficient condition for the smoothness assumption (Assumption 3.3). This is equivalent to that $\left\|\mathbf{L}_m^{-\frac{1}{2}}\mathbf{H}\mathbf{L}_m^{-\frac{1}{2}}\right\|_2 \leq 1/G_2$. Thus actually $\mathbf{L}_m$ can be taken as the optimal solution to an optimization problem:

$$\min_{\mathbf{L}\in\mathbb{R}_+^d}\left\|\mathbf{L}\right\|_1 \quad \text{s.t.} \quad \left\|\mathbf{L}_m^{-\frac{1}{2}}\mathbf{H}\mathbf{L}_m^{-\frac{1}{2}}\right\|_2 \leq \frac{1}{G_2},$$

which shares a similar form with the optimal diagonal preconditioner problem in solving linear systems, where an optimal preconditioner for a fixed matrix $\mathbf{H}$ can result in much faster convergence as pointed out in Qu et al. (2022). As an example, when $\mathbf{H}$ is diagonally dominant, we can choose $\mathbf{L}_m = \text{diagonal}(2G_2\mathbf{H})$ which takes the diagonal entries. Then we have $\left\|\mathbf{L}\right\|_\infty = 2G_2\mathbf{H}_{1,1}$ and $\left\|\mathbf{L}\right\|_1 = 2G_2\text{tr}(\mathbf{H})$. In this case, based on the heavy-tailed assumption (Assumption B.2), we have

$$R_2 = \frac{\phi(\mathbf{L})}{\phi(D)} = \frac{\text{tr}(\mathbf{H})D_\infty^2}{\mathbf{H}_{1,1}D_2^2} = \frac{\text{tr}(\mathbf{H})}{d\mathbf{H}_{1,1}} \propto \frac{\sum_{j=1}^{d}\exp(-\tau j)}{d\exp(-\frac{1}{2}\tau)} \leq \frac{1}{1-\exp\left(-\tau\right)}\cdot\frac{1}{d}.$$

When $\tau$ is a mild constant, it should be clear that $R_2$ can be close to $1/d$ and $R_2 \ll 1$.

To conclude, in this concrete example, we address that $R_1$ can be close to $1/\sqrt{d}$, $R_2$ can be close to $1/d$ and both $R_1, R_2 \ll 1$, showing that AdaGrad has better dimensional dependence than SGD and AdaGrad and the larger gap between bias terms than that of variance terms.

## C PROOF PRELIMINARIES

**Notations.**  In the appendix, we define

$$\mathbf{n}_t \triangleq \frac{1}{M} \sum_{\xi \in \mathcal{B}_t} \nabla_{\mathbf{w}} f(\mathbf{w}_t; \xi) - \nabla f(\mathbf{w}_t) = \frac{1}{M} \sum_{\xi \in \mathcal{B}_t} \mathbf{n}(\mathbf{w}_t; \xi) \tag{11}$$

to note the gradient noise at iteration $t$. We further use $\nabla f_{t,j}$, $\mathbf{g}_{t.j}$ and $\mathbf{n}_{t,j}$ to denote the $j$-th coordinate of $\nabla f(\mathbf{w}_t)$, $\mathbf{g}_t$ and $\mathbf{n}_t$, separately.

**Lemma C.1** (Projection). *Suppose $\mathcal{W} \subseteq \mathbb{R}^d$ is a closed convex set and $\mathbf{\Lambda} \in \mathbb{R}^{d \times d}$ is symmetric and positive definite. Then for $\mathbf{w} \in \mathbb{R}^d$, $\bar{\mathbf{w}} \in \Pi_C^{\mathbf{\Lambda}}[\mathbf{w}]$ if and only if for all $\mathbf{z} \in \mathcal{W}$,*

$$\frac{1}{2} \|\mathbf{w} - \mathbf{z}\|_{\mathbf{\Lambda}}^2 \geq \frac{1}{2} \|\mathbf{w} - \bar{\mathbf{w}}\|_{\mathbf{\Lambda}}^2 \tag{12}$$

*or equivalently, for all $\mathbf{z} \in \mathcal{W}$,*

$$\langle \mathbf{z} - \bar{\mathbf{w}}, \mathbf{\Lambda}(\mathbf{w} - \bar{\mathbf{w}}) \rangle \leq 0. \tag{13}$$

*Proof.* Equation (12) simply follows the definition. To prove (13), take any $\mathbf{z} \in \mathcal{W}$ and $\alpha \in (0, 1)$ we know that $\alpha \mathbf{z} + (1 - \alpha)\bar{\mathbf{w}} \in \mathcal{W}$ as $\mathcal{W}$ is a convex set. Hence by (12), we have

$$\frac{1}{2} \|\mathbf{w} - \bar{\mathbf{w}}\|_{\mathbf{\Lambda}}^2 \leq \frac{1}{2} \|\mathbf{w} - [\alpha \mathbf{z} + (1 - \alpha)\bar{\mathbf{w}}]\|_{\mathbf{\Lambda}}^2$$

$$\iff \quad \frac{1}{2} \|\mathbf{w} - \bar{\mathbf{w}}\|_{\mathbf{\Lambda}}^2 \leq \frac{1}{2} \|(\mathbf{w} - \bar{\mathbf{w}}) - \alpha(\mathbf{z} - \bar{\mathbf{w}})\|_{\mathbf{\Lambda}}^2$$

$$\iff \quad 0 \leq \frac{\alpha^2}{2} \|\mathbf{z} - \bar{\mathbf{w}}\|_{\mathbf{\Lambda}}^2 - \alpha \langle \mathbf{z} - \bar{\mathbf{w}}, \mathbf{\Lambda}(\mathbf{w} - \bar{\mathbf{w}}) \rangle$$

$$\iff \quad \langle \mathbf{z} - \bar{\mathbf{w}}, \mathbf{\Lambda}(\mathbf{w} - \bar{\mathbf{w}}) \rangle \leq \frac{\alpha}{2} \|\mathbf{z} - \bar{\mathbf{w}}\|_{\mathbf{\Lambda}}^2 .$$

As $\alpha$ can be arbitrarily close to 0, we have that (13) holds. $\square$

With the definition of (11), we can obtain the following lemma describing the variance reduced by batch size $M$.

**Lemma C.2** (Variance reduced by batch size). *Under Assumption 3.4, 3.5, it holds that*

$$\mathbb{E}_t \left[ \mathbf{n}_{t,j}^2 \right] \leq \frac{\boldsymbol{\sigma}_j^2}{M}. \tag{14}$$

*Proof.* It holds that

$$\mathbb{E}_t \left[ \mathbf{n}_{t,j}^2 \right] = \mathbb{E}_t \left[ \left( \frac{1}{M} \sum_{\xi \in \mathcal{B}_t} \mathbf{n}_j(\mathbf{w}_t; \xi) \right)^2 \right] \overset{(4)}{=} \frac{1}{M^2} \sum_{\xi \in \mathcal{B}_t} \mathbb{E}_t \left[ (\mathbf{n}_j(\mathbf{w}_t; \xi))^2 \right] \overset{(5)}{\leq} \frac{\boldsymbol{\sigma}_j^2}{M},$$

where the second equality is based on the independence of $\mathbf{n}(\mathbf{w}_t; \xi)$. $\square$

### C.1 PROPERTIES OF ASSUMPTION 5.1

The following is the descent lemma based on Assumption 5.1. We basically follow similar proof techniques to Zhang et al. (2019; 2020a); Crawshaw et al. (2022) but obtain results because of the difference in assumption.

**Lemma C.3** (Descent lemma). *Under Assumption 5.1, for all $\mathbf{w}, \mathbf{w}' \in \mathcal{W}$ such that $\|\mathbf{w} - \mathbf{w}'\|_{\mathbf{L}_1} \leq \sqrt{d}$, it holds that*

$$f(\mathbf{w}') \leq f(\mathbf{w}) + \langle \nabla f(\mathbf{w}), \mathbf{w}' - \mathbf{w} \rangle + \frac{1}{2} \sum_{j=1}^{d} (\mathbf{L}_{0,j} + \mathbf{L}_{1,j} \partial_j f(\mathbf{w})) |\mathbf{w}' - \mathbf{w}|_j^2, \tag{15}$$

*where $|\mathbf{w}' - \mathbf{w}|_j$ denotes the absolute value of the $j$-th coordinate of $\mathbf{w}' - \mathbf{w}$.*

---

**Algorithm 2** SGD

1: **Input:** $\mathbf{w}_0 \in \mathbb{R}^d$, $\{\eta_t\}_{t=0}^{T-1} \in \mathbb{R}$, and batch size $M \in \mathbb{N}$
2: **for** $t = 0$ **to** $T - 1$ **do**
3:     Sample mini-batch $\mathcal{B}_t$ with $|\mathcal{B}_t| \equiv M$ uniformly
4:     $\mathbf{g}_t = \frac{1}{M} \sum_{\xi \in \mathcal{B}_t} \nabla_{\mathbf{w}} f(\mathbf{w}_t; \xi)$
5:     **Option I:** $\mathbf{w}_{t+1} = \Pi_{\mathcal{W}}^{\mathbf{I}_d}(\mathbf{w}_t - \eta_t \mathbf{g}_t)$
6:     **Option II:** $\mathbf{w}_{t+1} = \mathbf{w}_t - \eta_t \mathbf{g}_t$
7: **end for**
8: **Output:** $1/T \sum_{t=0}^{T-1} \mathbf{w}_t$

---

**Algorithm 3** Adagrad-Norm

1: **Input:** $\mathbf{w}_0 \in \mathbb{R}^d$, $\{\eta_t\}_{t=0}^{T-1} \in \mathbb{R}$, $\epsilon \in \mathbb{R}$, and batch size $M \in \mathbb{N}$
2: Initialize $v_{-1} = \epsilon^2$
3: **for** $t = 0$ **to** $T - 1$ **do**
4:     Sample mini-batch $\mathcal{B}_t$ with $|\mathcal{B}_t| \equiv M$ uniformly
5:     $\mathbf{g}_t = \frac{1}{M} \sum_{\xi \in \mathcal{B}_t} \nabla_{\mathbf{w}} f(\mathbf{w}_t; \xi)$
6:     $v_t = v_{t-1} + \|\mathbf{g}_t\|_2^2$
7:     $b_t = \sqrt{v_t}$
8:     **Option I:** $\mathbf{w}_{t+1} = \Pi_{\mathcal{W}}^{\mathbf{\Lambda}_t}(\mathbf{w}_t - \frac{\eta_t}{b_t} \mathbf{g}_t)$
9:     **Option II:** $\mathbf{w}_{t+1} = \mathbf{w}_t - \frac{\eta_t}{b_t} \mathbf{g}_t$
10: **end for**
11: **Output:** $1/T \sum_{t=0}^{T-1} \mathbf{w}_t$

---

*Proof.* Denote $\mathbf{w}_u = u\mathbf{w}' + (1-u)\mathbf{w}$ with $u \in [0, 1]$. Based on the Taylor expansion, we have

$$
\begin{aligned}
f(\mathbf{w}') =& f(\mathbf{w}) + \int_0^1 \langle \nabla f(\mathbf{w}_u), \mathbf{w}' - \mathbf{w} \rangle \, \mathrm{d}u \\
=& f(\mathbf{w}) + \langle \nabla f(\mathbf{w}), \mathbf{w}' - \mathbf{w} \rangle + \int_0^1 \langle \nabla f(\mathbf{w}_u) - \nabla f(\mathbf{w}), \mathbf{w}' - \mathbf{w} \rangle \, \mathrm{d}u \\
=& f(\mathbf{w}) + \langle \nabla f(\mathbf{w}), \mathbf{w}' - \mathbf{w} \rangle + \int_0^1 \sum_{j=1}^d (\partial_j f(\mathbf{w}_u) - \partial_j f(\mathbf{w}))(\mathbf{w}' - \mathbf{w})_j \mathrm{d}u \\
\leq& f(\mathbf{w}) + \langle \nabla f(\mathbf{w}), \mathbf{w}' - \mathbf{w} \rangle \\
& + \int_0^1 \sqrt{\sum_{j=1}^d \frac{|\partial_j f(\mathbf{w}_u) - \partial_j f(\mathbf{w})|^2}{(\mathbf{L}_{0,j} + \mathbf{L}_{1,j}|\partial_j f(\mathbf{w})|)}} \cdot \sqrt{\sum_{j=1}^d (\mathbf{L}_{0,j} + \mathbf{L}_{1,j}|\partial_j f(\mathbf{w})|) |\mathbf{w}_u - \mathbf{w}|_j^2} \mathrm{d}u \\
\overset{(3)}{\leq}& f(\mathbf{w}) + \langle \nabla f(\mathbf{w}), \mathbf{w}' - \mathbf{w} \rangle + \int_0^1 \sum_{j=1}^d (\mathbf{L}_{0,j} + \mathbf{L}_{1,j}|\partial_j f(\mathbf{w})|) |\mathbf{w}_u - \mathbf{w}|_j^2 \, \mathrm{d}u \\
=& f(\mathbf{w}) + \langle \nabla f(\mathbf{w}), \mathbf{w}' - \mathbf{w} \rangle + \sum_{j=1}^d (\mathbf{L}_{0,j} + \mathbf{L}_{1,j}|\partial_j f(\mathbf{w})|) |\mathbf{w}' - \mathbf{w}|_j^2 \cdot \int_0^1 u \mathrm{d}u \\
=& f(\mathbf{w}) + \langle \nabla f(\mathbf{w}), \mathbf{w}' - \mathbf{w} \rangle + \frac{1}{2} \sum_{j=1}^d (\mathbf{L}_{0,j} + \mathbf{L}_{1,j}|\partial_j f(\mathbf{w})|) |\mathbf{w}' - \mathbf{w}|_j^2,
\end{aligned}
$$

where we use Cauchy-Schwarz inequality in the first inequality. $\square$

## D  STANDARD CONVERGENCE OF SGD

**Theorem D.1** (Convex convergence of SGD). *Under Assumption 3.1, 3.3, 3.4, 3.5, for the sequence $\{\mathbf{w}_t\}_{t=1}^T$ generated by SGD (Algorithm 2 with option I), if we appropriately take step size $\eta_t =$*

$$\min\left\{\frac{1}{\|\mathbf{L}\|_\infty}, \sqrt{\frac{D_2^2 M}{2\|\boldsymbol{\sigma}\|_2^2(t+1)}}\right\}, \text{ it holds that for } \bar{\mathbf{w}}_T \triangleq 1/T\sum_{t=0}^{T-1}\mathbf{w}_t,$$

$$\mathbb{E}\left[f(\bar{\mathbf{w}}_T) - f(\mathbf{w}_*)\right] = \mathcal{O}\left(\frac{D_2\|\boldsymbol{\sigma}\|_2}{\sqrt{MT}} + \frac{\|\mathbf{L}\|_\infty D_2^2}{T}\right).$$

*Remark* D.2. Here we consider a decaying step size schedule $\eta_t = \mathcal{O}\left(1/\sqrt{t}\right)$ for SGD, which is commonly used and usually more efficient than constant step size. Note that under the same settings, the convex convergence of SGD with the same step size schedule can also depend on just $\|\mathbf{w}_0 - \mathbf{w}_*\|_2$ instead of $D_2$, if we use a slightly different proof approach compared to Theorem D.1. One can refer to Theorem 5.7 in Garrigos & Gower (2023) for this different approach. While it can obtain convergence with dependence on $\|w_0 - w_*\|_2$ instead of $D_2$, it suffers from an additional logarithmic term in the bound when using step size $\eta_t = \mathcal{O}\left(1/\sqrt{t}\right)$. Therefore, we think each of these two approaches has its own merits and we take the current approach for discussion.

**Theorem D.3** (Nonconvex convergence of SGD). *Under Assumption 3.2, 3.3, 3.4, 3.5, for the sequence $\{\mathbf{w}_t\}_{t=1}^T$ generated by SGD (Algorithm 2 with option II), if we appropriately take step size $\eta_t \equiv \eta = \left\{\frac{1}{2\|\mathbf{L}\|_\infty}, \sqrt{\frac{2(f(\mathbf{w}_0)-f^*)M}{\|\mathbf{L}\|_\infty\|\boldsymbol{\sigma}\|_2^2 T}}\right\}$, it holds that*

$$\frac{1}{T}\sum_{t=0}^{T-1}\mathbb{E}\left[\|\nabla f(\mathbf{w}_t)\|_2^2\right] = \mathcal{O}\left(\frac{\sqrt{\|\mathbf{L}\|_\infty(f(\mathbf{w}_0)-f^*)}\|\boldsymbol{\sigma}\|_2}{\sqrt{MT}} + \frac{\|\mathbf{L}\|_\infty(f(\mathbf{w}_0)-f^*)}{T}\right).$$

Theorem D.1 and D.3 are standard results of the convergence of SGD under $\|\mathbf{L}\|_\infty$-smooth settings. Similar results have been extensively studied (e.g. (Orabona, 2019; Garrigos & Gower, 2023; Ghadimi & Lan, 2013; Bernstein et al., 2018a)). We also include the proof below for completeness.

*Proof of Theorem D.1.* For $\mathbf{w}_* \in \mathcal{W}$, it holds that

$$\mathbb{E}_t\left[\|\mathbf{w}_{t+1} - \mathbf{w}_*\|_2^2\right] \overset{(12)}{\leq} \mathbb{E}_t\left[\|\mathbf{w}_t - \eta_t\mathbf{g}_t - \mathbf{w}_*\|_2^2\right]$$

$$\leq \|\mathbf{w}_t - \mathbf{w}_*\|_2^2 - 2\eta_t\mathbb{E}_t\left[\langle\mathbf{g}_t, \mathbf{w}_t - \mathbf{w}_*\rangle\right] + \eta_t^2\mathbb{E}_t\left[\|\mathbf{g}_t\|_2^2\right]$$

$$= \|\mathbf{w}_t - \mathbf{w}_*\|_2^2 - 2\eta_t\langle\nabla f(\mathbf{w}_t), \mathbf{w}_t - \mathbf{w}_*\rangle + \eta_t^2\|\nabla f(\mathbf{w}_t)\|_2^2 + \eta_t^2\mathbb{E}_t\left[\|\mathbf{n}_t\|_2^2\right],$$

where $\mathbf{n}_t = \mathbf{g}_t - \nabla f(\mathbf{w}_t)$. With convexity and standard smoothness of $f(\cdot)$, we have

$$\langle\nabla f(\mathbf{w}_t), \mathbf{w}_t - \mathbf{w}_*\rangle \geq f(\mathbf{w}_t) - f(\mathbf{w}_*) \quad \text{and} \quad \langle\nabla f(\mathbf{w}_t), \mathbf{w}_t - \mathbf{w}_*\rangle \geq \frac{1}{\|\mathbf{L}\|_\infty}\|\nabla f(\mathbf{w}_t)\|_2^2.$$

Then after rearranging, it holds that

$$(2 - \eta_t\|\mathbf{L}\|_\infty)(f(\mathbf{w}_t) - f(\mathbf{w}_*)) \leq \frac{1}{\eta_t}\|\mathbf{w}_t - \mathbf{w}_*\|_2^2 - \frac{1}{\eta_t}\mathbb{E}_t\left[\|\mathbf{w}_{t+1} - \mathbf{w}_*\|_2^2\right] + \eta_t\mathbb{E}_t\left[\|\mathbf{n}_t\|_2^2\right].$$

If we take summation and full expectation with $\eta_t \leq 1/L$ and $\eta_t \leq \eta_{t+1}$ for $t = 1, \cdots, T$, it holds that

$$\sum_{t=0}^{T-1}\mathbb{E}\left[f(\mathbf{w}_t) - f(\mathbf{w}_*)\right] \leq \sum_{t=0}^{T-1}\mathbb{E}\left[\frac{1}{\eta_t}\|\mathbf{w}_t - \mathbf{w}_*\|_2^2 - \frac{1}{\eta_t}\|\mathbf{w}_{t+1} - \mathbf{w}_*\|_2^2\right] + \sum_{t=0}^{T-1}\eta_t\mathbb{E}\left[\|\mathbf{n}_t\|_2^2\right]$$

$$\overset{(14)}{\leq} \sum_{t=0}^{T-1}\mathbb{E}\left[\frac{1}{\eta_t}\|\mathbf{w}_t - \mathbf{w}_*\|_2^2 - \frac{1}{\eta_t}\|\mathbf{w}_{t+1} - \mathbf{w}_*\|_2^2\right] + \sum_{t=0}^{T-1}\frac{\eta_t\|\boldsymbol{\sigma}\|_2^2}{M}$$

$$= \frac{1}{\eta_0}\|\mathbf{w}_0 - \mathbf{w}_*\|_2^2 - \frac{1}{\eta_{T-1}}\mathbb{E}\left[\|\mathbf{w}_T - \mathbf{w}_*\|_2^2\right] + \sum_{t=1}^{T-1}\left(\frac{1}{\eta_t} - \frac{1}{\eta_{t-1}}\right)\mathbb{E}\left[\|\mathbf{w}_t - \mathbf{w}_*\|_2^2\right]$$

$$+ \sum_{t=0}^{T-1}\frac{\eta_t\|\boldsymbol{\sigma}\|_2^2}{M}$$

$$\overset{(2)}{\leq} \frac{D_2^2}{\eta_0} + \sum_{t=1}^{T-1}\left(\frac{1}{\eta_t} - \frac{1}{\eta_{t-1}}\right)D_2^2 + \sum_{t=0}^{T-1}\frac{\eta_t\|\boldsymbol{\sigma}\|_2^2}{M}$$

$$= \frac{D_2^2}{\eta_{T-1}} + \sum_{t=0}^{T-1}\frac{\eta_t\|\boldsymbol{\sigma}\|_2^2}{M}.$$

Therefore, by having both sides of the inequality divided by $T$ and making use of the convexity, we conclude the proof that

$$\mathbb{E}\left[f(\bar{\mathbf{w}}_T) - f(\mathbf{w}_*)\right] \overset{\text{Asm. 3.1}}{\leq} \frac{1}{T}\sum_{t=0}^{T-1}\mathbb{E}\left[f(\mathbf{w}_t) - f(\mathbf{w}_*)\right] \leq \frac{D_2^2}{\eta_{T-1}T} + \frac{\|\boldsymbol{\sigma}\|_2^2}{MT}\sum_{t=0}^{T-1}\eta_t,$$

where $\bar{\mathbf{w}}_T = 1/T\sum_{t=0}^{T-1}\mathbf{w}_t$. Then by taking

$$\eta_t = \min\left\{\frac{1}{\|\mathbf{L}\|_\infty}, \sqrt{p\frac{D_2^2 M}{\sigma_2^2(t+1)}}\right\},$$

where $p$ is a constant, we can obtain that

$$\mathbb{E}\left[f(\bar{\mathbf{w}}_T) - f(\mathbf{w}_*)\right] \leq \frac{D_2^2}{\eta_{T-1}T} + \frac{\|\boldsymbol{\sigma}\|_2^2}{MT}\sum_{t=0}^{t-1}\eta_t$$

$$\leq \frac{\|\mathbf{L}\|_\infty D_2^2}{T} + \sqrt{\frac{D_2^2\|\boldsymbol{\sigma}\|_2^2}{MT}}\cdot\sqrt{\frac{1}{p}} + \sqrt{\frac{D_2^2\|\boldsymbol{\sigma}\|_2^2}{MT}}\cdot 2\sqrt{p},$$

where the second inequality holds as

$$\sum_{t=0}^{T-1}\frac{1}{\sqrt{t+1}} \leq 2\sum_{t=0}^{T-1}\frac{1}{\sqrt{t+1}+\sqrt{t}} = 2\sum_{t=0}^{T-1}\frac{\sqrt{t+1}-\sqrt{t}}{t+1-t} = 2\sum_{t=0}^{T-1}\sqrt{t+1} - \sqrt{t} = 2\sqrt{T}.$$

Thus by taking $p = \frac{1}{2}$, we finish the proof. $\qquad\square$

*Proof of Theorem D.3.* Based on the smoothness condition, it holds that

$$f(\mathbf{w}_{t+1}) \leq f(\mathbf{w}_t) - \langle\nabla f(\mathbf{w}_t), \mathbf{w}_{t+1} - \mathbf{w}_t\rangle + \frac{\|\mathbf{L}\|_\infty}{2}\|\mathbf{w}_{t+1} - \mathbf{w}_t\|_2^2$$

$$= f(\mathbf{w}_t) - \eta_t\langle\nabla f(\mathbf{w}_t), \mathbf{g}_t\rangle + \frac{\|\mathbf{L}\|_\infty\eta_t^2}{2}\|\mathbf{g}_t\|_2^2$$

$$\leq f(\mathbf{w}_t) - \eta_t\langle\nabla f(\mathbf{w}_t), \mathbf{g}_t\rangle + \|\mathbf{L}\|_\infty\eta_t^2\|\nabla f(\mathbf{w}_t)\|_2^2 + \|\mathbf{L}\|_\infty\eta_t^2\|\mathbf{n}_t\|_2^2.$$

By taking expectation, we have

$$\mathbb{E}\left[f(\mathbf{w}_{t+1})\right] \leq \mathbb{E}\left[f(\mathbf{w}_t)\right] - \eta_t\mathbb{E}\left[\langle\nabla f(\mathbf{w}_t), \mathbf{g}_t\rangle\right] + \frac{\|\mathbf{L}\|_\infty\eta_t^2}{2}\mathbb{E}\left[\|\mathbf{g}_t\|_2^2\right]$$

$$= \mathbb{E}\left[f(\mathbf{w}_t)\right] - \eta_t\mathbb{E}\left[\|\nabla f(\mathbf{w}_t)\|_2^2\right] + \frac{\|\mathbf{L}\|_\infty\eta_t^2}{2}\mathbb{E}\left[\|\nabla f(\mathbf{w}_t)\|_2^2\right] + \frac{\|\mathbf{L}\|_\infty\eta_t^2}{2}\mathbb{E}\left[\|\mathbf{n}_t\|_2^2\right]$$

$$\overset{(14)}{\leq} \mathbb{E}\left[f(\mathbf{w}_t)\right] - \eta_t\mathbb{E}\left[\|\nabla f(\mathbf{w}_t)\|_2^2\right] + \frac{\|\mathbf{L}\|_\infty\eta_t^2}{2}\mathbb{E}\left[\|\nabla f(\mathbf{w}_t)\|_2^2\right] + \frac{\|\mathbf{L}\|_\infty\eta_t^2\|\boldsymbol{\sigma}\|_2^2}{2M}$$

$$\leq \mathbb{E}\left[f(\mathbf{w}_t)\right] - \frac{\eta_t}{2}\mathbb{E}\left[\|\nabla f(\mathbf{w}_t)\|_2^2\right] + \frac{\|\mathbf{L}\|_\infty\eta_t^2\|\boldsymbol{\sigma}\|_2^2}{2M},$$

where $\mathbf{n}_t = \nabla f(\mathbf{w}_t) - \mathbf{g}_t$ and the last inequality holds as we take $\eta_t \leq 1/\|\mathbf{L}\|_\infty$. As $\eta_t \equiv \eta$, we have after rearranging that

$$\frac{1}{T}\sum_{t=0}^{T-1}\mathbb{E}\left[\|\nabla f(\mathbf{w}_t)\|_2^2\right] \leq \frac{2\mathbb{E}[f(\mathbf{w}_0) - f(\mathbf{w}_T)]}{\eta T} + \frac{\|\mathbf{L}\|_\infty\eta\|\boldsymbol{\sigma}\|_2^2}{M}$$

$$\leq \frac{2(f(\mathbf{w}_0) - f^*)}{\eta T} + \frac{\|\mathbf{L}\|_\infty\eta\|\boldsymbol{\sigma}\|_2^2}{M}.$$

Then by taking $\eta = \min\left\{\frac{1}{\|\mathbf{L}\|_\infty}, \sqrt{\frac{(f(\mathbf{w}_0)-f^*)M}{\|\mathbf{L}\|_\infty\|\boldsymbol{\sigma}\|_2^2 T}}\right\}$, we finish the proof. $\qquad\square$

# E PROOF OF THEOREM 4.1

To prove Theorem 4.1, we first look at the standard AdaGrad convergence under the non-smooth convex stochastic optimization setting.

**Theorem E.1** (Convergence for convex nonsmooth AdaGrad). *Under Assumption 3.1, 3.4, for the sequence $\{\mathbf{w}_t\}_{t=1}^{T}$ generated by Algorithm 1 with constant step size $\eta_t \equiv \eta$, it holds that*

$$\sum_{t=0}^{T-1} \mathbb{E}[f(\mathbf{w}_t)] - f(\mathbf{w}_*) \le \frac{1}{\sqrt{2}} \left( \eta + \frac{D_\infty^2}{\eta} \right) \sum_{j=1}^{d} \mathbb{E} \left[ \sqrt{\sum_{t=0}^{T-1} \nabla f_{t,j}^2} + \sqrt{\sum_{t=0}^{T-1} \mathbf{n}_{t,j}^2} \right] + \frac{\epsilon D_2^2}{2\eta} \qquad (16)$$

*Proof.* This proof is a stochastic optimization version of the proof of AdaGrad in the online convex learning scheme Duchi et al. (2011); Streeter & McMahan (2010); Zhang (2023). We also include the proof here for completeness.

First, we give an important result that the gradient norm can be expressed as $\boldsymbol{\Lambda}_t - \boldsymbol{\Lambda}_{t-1}$.

$$\|\mathbf{g}_t\|_{\boldsymbol{\Lambda}_t^{-1}}^2 = \sum_{j=1}^{d} \frac{\mathbf{g}_{t,j}^2}{\lambda_{t,j}} \le \sum_{j=1}^{d} \frac{\mathbf{g}_{t,j}^2}{\lambda_{t,j} + \lambda_{t-1,j}} = \sum_{j=1}^{d} \frac{\lambda_{t,j}^2 - \lambda_{t-1,j}^2}{\lambda_{t,j} + \lambda_{t-1,j}} = \mathrm{tr}\left( \boldsymbol{\Lambda}_t - \boldsymbol{\Lambda}_{t-1} \right), \qquad (17)$$

where we note $\lambda_{t,j}$ is the $j$-th coordinate of $\boldsymbol{\Lambda}_t$. Then we start the proof.

$$
\begin{aligned}
\mathbb{E}_t \left[ \|\mathbf{w}_{t+1} - \mathbf{w}_*\|_{\boldsymbol{\Lambda}_t}^2 \right] &\overset{(12)}{\le} \mathbb{E} \left[ \|\mathbf{w}_t - \mathbf{w}_* - \eta_t \boldsymbol{\Lambda}_t^{-1} \mathbf{g}_t\|_{\boldsymbol{\Lambda}_t}^2 \right] \\
&= \mathbb{E}_t \left[ \|\mathbf{w}_t - \mathbf{w}_*\|_{\boldsymbol{\Lambda}_t}^2 - 2\eta_t \langle \mathbf{g}_t, \mathbf{w}_t - \mathbf{w}_* \rangle + \eta_t^2 \|\mathbf{g}_t\|_{\boldsymbol{\Lambda}_t^{-1}}^2 \right] \\
&= \mathbb{E}_t \left[ \|\mathbf{w}_t - \mathbf{w}_*\|_{\boldsymbol{\Lambda}_t}^2 \right] - 2\eta_t \langle \nabla f(\mathbf{w}_t), \mathbf{w}_t - \mathbf{w}_* \rangle + \mathbb{E}_t \left[ \eta_t^2 \|\mathbf{g}_t\|_{\boldsymbol{\Lambda}_t^{-1}}^2 \right] \\
&\overset{(17)}{\le} \mathbb{E}_t \left[ \|\mathbf{w}_t - \mathbf{w}_*\|_{\boldsymbol{\Lambda}_t}^2 \right] - 2\eta_t \langle \nabla f(\mathbf{w}_t), \mathbf{w}_t - \mathbf{w}_* \rangle + \mathbb{E}_t \left[ \eta_t^2 \mathrm{tr}\left( \boldsymbol{\Lambda}_t - \boldsymbol{\Lambda}_{t-1} \right) \right] \\
&\le \|\mathbf{w}_t - \mathbf{w}_*\|_{\boldsymbol{\Lambda}_{t-1}}^2 - 2\eta_t \left( f(\mathbf{w}_t) - f(\mathbf{w}_*) \right) + \mathbb{E}_t \left[ \eta_t^2 \mathrm{tr}\left( \boldsymbol{\Lambda}_t - \boldsymbol{\Lambda}_{t-1} \right) \right] \\
&\quad + \mathbb{E}_t \left[ \|\mathbf{w}_t - \mathbf{w}_*\|_{\boldsymbol{\Lambda}_t - \boldsymbol{\Lambda}_{t-1}}^2 \right] \\
&\overset{(2)}{\le} \|\mathbf{w}_t - \mathbf{w}_*\|_{\boldsymbol{\Lambda}_{t-1}}^2 - 2\eta_t \left( f(\mathbf{w}_t) - f(\mathbf{w}_*) \right) + \mathbb{E}_t \left[ \eta_t^2 \mathrm{tr}\left( \boldsymbol{\Lambda}_t - \boldsymbol{\Lambda}_{t-1} \right) \right] \\
&\quad + \mathbb{E}_t \left[ D_\infty^2 \mathrm{tr}\left( \boldsymbol{\Lambda}_t - \boldsymbol{\Lambda}_{t-1} \right) \right],
\end{aligned}
$$

where the third inequality holds as $f(\cdot)$ is convex. After taking $\eta_t = \eta$, summing up, taking full expectation and rearrangement, we can obtain that

$$
\begin{aligned}
\sum_{t=0}^{T-1} \mathbb{E}\left[ f(\mathbf{w}_t) - f(\mathbf{w}_*) \right] &\le \frac{1}{2\eta} \sum_{t=0}^{T-1} \mathbb{E} \left[ \|\mathbf{w}_t - \mathbf{w}_*\|_{\boldsymbol{\Lambda}_{t-1}}^2 - \|\mathbf{w}_{t+1} - \mathbf{w}_*\|_{\boldsymbol{\Lambda}_t}^2 \right] \\
&\quad + \left( \frac{\eta}{2} + \frac{D_\infty^2}{2\eta} \right) \sum_{t=0}^{T-1} \mathbb{E} \left[ \mathrm{tr}\left( \boldsymbol{\Lambda}_t - \boldsymbol{\Lambda}_{t-1} \right) \right] \\
&= \frac{\epsilon \|\mathbf{w}_0 - \mathbf{w}_*\|_2^2}{2\eta} - \frac{\|\mathbf{w}_T - \mathbf{w}_*\|_{\boldsymbol{\Lambda}_{T-1}}^2}{2\eta} + \left( \frac{\eta}{2} + \frac{D_\infty^2}{2\eta} \right) \mathbb{E} \left[ \mathrm{tr}\left( \boldsymbol{\Lambda}_{T-1} - \epsilon \mathbf{I}_d \right) \right] \\
&\overset{(2)}{\le} \frac{\epsilon D_2^2}{2\eta} + \left( \frac{\eta}{2} + \frac{D_\infty^2}{2\eta} \right) \sum_{j=1}^{d} \mathbb{E} \left[ \sqrt{\sum_{t=0}^{T-1} \mathbf{g}_{t,j}^2} \right] \\
&\le \frac{\epsilon D_2^2}{2\eta} + \sqrt{2} \left( \frac{\eta}{2} + \frac{D_\infty^2}{2\eta} \right) \sum_{j=1}^{d} \mathbb{E} \left[ \sqrt{\sum_{t=0}^{T-1} \nabla f_{t,j}^2} + \sqrt{\sum_{t=0}^{T-1} \mathbf{n}_{t,j}^2} \right],
\end{aligned}
$$

where we take $\boldsymbol{\Lambda}_{-1} = \epsilon \mathbf{I}_d$ and the last inequality holds as $\sqrt{x+y} \le \sqrt{x} + \sqrt{y}$ for all $x, y \ge 0$. $\qquad \square$

Then we consider giving a bound on the noise summation term.

**Lemma E.2** (variance bound). *Under Assumption 3.4 and 3.5, it holds that*

$$\frac{1}{T} \sum_{j=1}^{d} \mathbb{E} \left[ \sqrt{\sum_{t=0}^{T-1} \mathbf{n}_{t,j}^2} \right] \le \frac{\|\boldsymbol{\sigma}\|_1}{\sqrt{MT}}.$$

*Proof.* It holds that

$$\sum_{j=1}^{d} \mathbb{E} \left[ \sqrt{\sum_{t=0}^{T-1} \mathbf{n}_{t,j}^2} \right] \le \sum_{j=1}^{d} \mathbb{E} \left[ \sqrt{\sum_{t=0}^{T-1} \mathbb{E}_t \left[ \mathbf{n}_{t,j}^2 \right]} \right] \overset{(14)}{\le} \sum_{j=1}^{d} \sqrt{\sum_{t=0}^{T-1} \frac{\boldsymbol{\sigma}_j^2}{M}} = \frac{\sqrt{T} \|\boldsymbol{\sigma}\|_1}{\sqrt{M}},$$

where the first inequality holds as Jensen's inequality. $\square$

Then we are ready to prove Theorem 4.1.

*Proof of Theorem 4.1.* From Theorem E.1, we can obtain that

$$\sum_{t=0}^{T-1} \mathbb{E}[f(\mathbf{w}_t)] - f(\mathbf{w}_*) \overset{(16)}{\le} \frac{1}{\sqrt{2}} \left( \eta + \frac{D_\infty^2}{\eta} \right) \sum_{j=1}^{d} \mathbb{E} \left[ \sqrt{\sum_{t=0}^{T-1} \nabla f_{t,j}^2} + \sqrt{\sum_{t=0}^{T-1} \mathbf{n}_{t,j}^2} \right] + \frac{\epsilon D_2^2}{2\eta},$$

and in Lemma E.2 we bound the noise term. Then we consider the bias term. It holds that

$$\sum_{j=1}^{d} \mathbb{E} \left[ \sqrt{\sum_{t=0}^{T-1} \nabla f_{t,j}^2} \right] \overset{(30)}{\le} \mathbb{E} \left[ \sqrt{\|\mathbf{L}\|_1 \sum_{j=1}^{d} \sum_{t=0}^{T-1} \frac{\nabla f_{t,j}^2}{L_j}} \right]$$

$$\le \sqrt{\|\mathbf{L}\|_1 \sum_{t=0}^{T-1} \mathbb{E} \left[ \sum_{j=1}^{d} \frac{\nabla f_{t,j}^2}{L_j} \right]} = \sqrt{\|\mathbf{L}\|_1 \sum_{t=0}^{T-1} \mathbb{E} \left[ \|\nabla f(\mathbf{w}_t)\|_{\mathbf{L}^{-1}}^2 \right]},$$

where the first inequality uses Lemma G.1 and the second inequality holds as the Jensen's inequality. Moreover, based on the smoothness assumption (Assumption 3.3), we can obtain that

$$\frac{1}{2T} \sum_{t=0}^{T-1} \mathbb{E} \left[ \|\nabla f(\mathbf{w}_t)\|_{\mathbf{L}^{-1}}^2 \right] \le \frac{1}{T} \sum_{t=0}^{T-1} \mathbb{E} \left[ f(\mathbf{w}_t) - f(\mathbf{w}_*) \right].$$

Thus if we denote $A = 1/T \sum_{t=0}^{T-1} \mathbb{E} \left[ f(\mathbf{w}_t) - f(\mathbf{w}_*) \right]$, then combining (16), there is a simplified expression that

$$A - CC_1 \sqrt{A} - CC_0 \le 0,$$

where from Theorem E.1 and Lemma E.2,

$$C = \frac{1}{\sqrt{2}} \left( \eta + \frac{D_\infty^2}{\eta} \right), \quad C_1 = \sqrt{\frac{2 \|\mathbf{L}\|_1}{T}}, \quad C_0 = \frac{\|\boldsymbol{\sigma}\|_1}{\sqrt{MT}} + \frac{\epsilon D_2^2}{2\eta TC}.$$

Then we can solve this inequality by conducting simple derivation that

$$\sqrt{A} \le \frac{1}{2} \left[ CC_1 + \sqrt{C^2 C_1^2 + 4CC_0} \right]$$

$$\implies A \le \frac{1}{2} \left[ C^2 C_1^2 + \left( C^2 C_1^2 + 4CC_0 \right) \right]$$

$$= C^2 C_1^2 + 2CC_0.$$

If we replace $C, C_0$ and $C_1$ by their corresponding value, we can obtain that

$$\frac{1}{T} \sum_{t=0}^{T-1} \mathbb{E} \left[ f(\mathbf{w}_t) - f(\mathbf{w}_*) \right] \le \frac{1}{2} \left( \eta + \frac{D_\infty^2}{\eta} \right)^2 \frac{2 \|\mathbf{L}\|_1}{T} + \frac{1}{\sqrt{2}} \left( \eta + \frac{D_\infty^2}{\eta} \right) \frac{\|\boldsymbol{\sigma}\|_1}{\sqrt{MT}} + \frac{\epsilon D_2^2}{\eta T}$$

If we further take the optimal step size $\eta = D_\infty$ based on this bound, we can obtain that

$$\frac{1}{T} \sum_{t=0}^{T-1} \mathbb{E} \left[ f(\mathbf{w}_t) - f(\mathbf{w}_*) \right] \le \frac{4 \|\mathbf{L}\|_1 D_\infty^2}{T} + \frac{\sqrt{2} D_\infty \|\boldsymbol{\sigma}\|_1}{\sqrt{MT}} + \frac{\epsilon D_2^2}{D_\infty T},$$

which concludes the proof. $\square$

### E.1    RELAXATION OF THE NOISE ASSUMPTION

Note that Assumption 3.5 assumes a universal bound on the noise. However, similar to a line of recent works on SGD (Richtárik & Takác, 2020; Gower et al., 2019; 2021; Khaled & Richtárik, 2020), Theorem 4.1 can also be extended to the settings that we only assume upper bound of noise variance on $\mathbf{w}_*$ as Assumption E.3, with Assumption 3.3 replaced by an assumption that aligns with the expected smoothness (Assumption E.4). Based on this setting, our analysis can be well adapted to the interpolation or overparameterized scheme.

**Assumption E.3** (Anisotropic noise on $\mathbf{w}_*$). There exists a positive vector $\boldsymbol{\sigma}_* = [\boldsymbol{\sigma}_{*,1}, \ldots, \boldsymbol{\sigma}_{*,d}] \in \mathbb{R}^d$ such that for all $\mathbf{w}_* \in \mathcal{W}$ that are minimum of $f(\cdot)$, it holds that

$$\mathbb{E}\left[\mathbf{n}_j(\mathbf{w}_*; \xi)^2\right] \leq \boldsymbol{\sigma}_{*,j}^2 \quad \text{for all} \quad j \in [d]. \tag{18}$$

**Assumption E.4** (Expected anisotropic smoothness). There exists a positive vector $\mathbf{L}_* = [\mathbf{L}_{*,1}, \ldots, \mathbf{L}_{*,d}] \in \mathbb{R}_+^d$ such that $f(\cdot)$ is $\mathbf{L}_*$-expected smooth, namely, for $\mathbf{w} \in \mathcal{W}$, $\mathbf{g}(\cdot)$ is the stochastic gradient oracle and all $\mathbf{w}_* \in \mathcal{W}$ that are the minimum of $f(\cdot)$, it holds that

$$\frac{1}{2}\mathbb{E}\left[\|\mathbf{g}(\mathbf{w}) - \mathbf{g}(\mathbf{w}_*)\|_{\mathbf{L}_*^{-1}}^2\right] \leq f(\mathbf{w}) - f(\mathbf{w}^*). \tag{19}$$

With these two assumptions replaced, we may rewrite the last step in Theorem E.1 such that

$$\sum_{j=1}^d \mathbb{E}\left[\sqrt{\sum_{t=0}^{T-1} \mathbf{g}_{t,j}^2}\right] \leq \sum_{j=1}^d \sqrt{\sum_{t=0}^{T-1} \mathbb{E}\left[\mathbf{g}_{t,j}^2\right]}$$

$$\leq \sum_{j=1}^d \sqrt{\sum_{t=0}^{T-1} \mathbb{E}\left[2\left|\mathbf{g}(\mathbf{w}_t) - \mathbf{g}(\mathbf{w}_*)\right|_j^2\right]} + \sum_{j=1}^d \sqrt{\sum_{t=0}^{T-1} \mathbb{E}\left[2\left|\mathbf{g}(\mathbf{w}_*)\right|_j^2\right]}$$

$$\overset{(30),(18)}{\leq} \sqrt{2\|\mathbf{L}_*\|_1 \sum_{t=0}^{T-1} \mathbb{E}\left[\|\mathbf{g}(\mathbf{w}_t) - \mathbf{g}(\mathbf{w}_*)\|_{\mathbf{L}_*^{-1}}^2\right]} + \|\boldsymbol{\sigma}_*\|_1 \sqrt{\frac{2T}{M}}$$

$$\overset{(19)}{\leq} \sqrt{4\|\mathbf{L}_*\|_1 \sum_{t=0}^{T-1} \mathbb{E}\left[f(\mathbf{w}_t) - f(\mathbf{w}_*)\right]} + \|\boldsymbol{\sigma}_*\|_1 \sqrt{\frac{2T}{M}}.$$

Then we can process the proof of Theorem 4.1 in the same manner, which gives the following convergence result:

$$\frac{1}{T}\sum_{t=0}^{T-1} \mathbb{E}\left[f(\mathbf{w}_t) - f(\mathbf{w}_*)\right] \leq \mathcal{O}\left(\frac{\|\mathbf{L}_*\|_1 D_\infty^2}{T} + \frac{D_\infty \|\boldsymbol{\sigma}_*\|_1}{\sqrt{MT}} + \frac{\epsilon D_2^2}{D_\infty T}\right).$$

Over-parameterized models usually result in interpolation of the data, which typically holds $\nabla_{\mathbf{w}} f(\mathbf{w}_*; \xi) = \nabla f(\mathbf{w}) = \mathbf{0}$, which corresponds to that $\boldsymbol{\sigma}_* = \mathbf{0}$. Therefore, we have $\mathcal{O}(1/T)$ convergence rate for AdaGrad under over-parameterization based on this relaxation of the noise assumption.

## F    PROOF OF THEOREM 5.3

In this section, we prove the convergence of AdaGrad in the nonconvex generalized smooth setting. Note that by setting $\mathbf{L}_1 = 0$, the proof below can be directly applied to prove Theorem 4.3. Let us first give a brief overview. Based on Assumption 3.3 and Lemma C.3, when we set $\eta_t \equiv \eta \leq 1/\|\mathbf{L}_1\|_\infty$, it holds that $\|\mathbf{w}_{t+1} - \mathbf{w}_t\|_{\mathbf{L}_1} \leq \sqrt{d}$ and

$$f(\mathbf{w}_{t+1}) \overset{(15)}{\leq} f(\mathbf{w}_t) + \langle\nabla f(\mathbf{w}_t), \mathbf{w}_{t+1} - \mathbf{w}_t\rangle + \frac{1}{2}\sum_{j=1}^d \left(\mathbf{L}_{0,j} + \mathbf{L}_{1,j}\left|\nabla f_{t,j}\right|\right)\left|\mathbf{w}_{t+1} - \mathbf{w}_t\right|_j^2$$

$$= f(\mathbf{w}_t) - \eta\langle\nabla f(\mathbf{w}_t), \boldsymbol{\Lambda}_t^{-1}\mathbf{g}_t\rangle + \frac{\eta^2}{2}\sum_{j=1}^d \left(\mathbf{L}_{0,j} + \mathbf{L}_{1,j}\left|\nabla f_{t,j}\right|\right)\frac{\mathbf{g}_{t,j}^2}{\lambda_{t,j}}.$$

Here a critical problem is it is nontrivial to straightforwardly transfer $\mathbb{E}_t[\langle \nabla f(\mathbf{w}_t), \mathbf{\Lambda}_t^{-1}\mathbf{g}_t\rangle]$ to $\mathbb{E}_t[\|\nabla f(\mathbf{w})\|]$ as both $\mathbf{\Lambda}_t$ and $\mathbf{g}_t$ is relevant with $\mathbf{g}_t$. To deal with this issue, we consider an auxiliary diagonal matrix $\tilde{\mathbf{\Lambda}}_t$ with each diagonal entry being

$$\tilde{\lambda}_{t,j}^2 = \lambda_{t-1,j}^2 + \mathbb{E}_t\left[\mathbf{g}_{t,j}^2\right] \tag{20}$$

for all $j \in [d]$. Note that this auxiliary sequence is the same as Défossez et al. (2020), but our proof technique is much different and obtains better results. Then we consider

$$-\left\langle \nabla f(\mathbf{w}_t), \mathbf{\Lambda}_t^{-1}\mathbf{g}_t\right\rangle = -\left\langle \nabla f(\mathbf{w}_t), \tilde{\mathbf{\Lambda}}_t^{-1}\mathbf{g}_t\right\rangle + \left\langle \nabla f(\mathbf{w}_t), \left(\tilde{\mathbf{\Lambda}}_t^{-1} - \mathbf{\Lambda}_t^{-1}\right)\mathbf{g}_t\right\rangle$$

and attempt to bound the second term, which is described in Lemma F.1.

Another problem is how to bound the additional terms introduced by the generalized smoothness, namely, $\sum_{j=1}^d \mathbf{L}_{1,j}|\nabla f_{t,j}|\frac{\mathbf{g}_{t,j}^2}{\lambda_{t,j}}$. We use the divide-and-conquer strategy to have this additional term resolved by the existing terms, as described in Lemma F.2.

With these issues solved, the final convergence property would be determined by

$$\sum_{t=0}^{T-1}\mathbb{E}\left[\left\langle \nabla f(\mathbf{w}_t), \tilde{\mathbf{\Lambda}}_t^{-1}\mathbf{g}_t\right\rangle\right],$$

which largely relies on $\mathbb{E}[\operatorname{tr}(\mathbf{\Lambda}_t)]$ that we bound in Lemma F.4. Note that Lemma F.4 considers a different main line and is important for the proof. Then we are ready to complete the proof of Theorem 5.3.

**Lemma F.1** (Bound of $\left\langle \nabla f(\mathbf{w}_t), \left(\tilde{\mathbf{\Lambda}}_t^{-1} - \mathbf{\Lambda}_t^{-1}\right)\mathbf{g}_t\right\rangle$). *Under the same setting as Theorem 5.3, if we take diagonal matrix $\tilde{\mathbf{\Lambda}}_t$ as defined in* (20), *it holds that*

$$\mathbb{E}_t\left[\left\langle \nabla f(\mathbf{w}_t), \left(\tilde{\mathbf{\Lambda}}_t^{-1} - \mathbf{\Lambda}_t^{-1}\right)\mathbf{g}_t\right\rangle\right] \le \sum_{j=1}^d 2\boldsymbol{\sigma}_j\mathbb{E}_t\left[\frac{\mathbf{g}_{t,j}^2}{\lambda_{t,j}^2}\right] + \frac{1}{2}\left\langle \nabla f(\mathbf{w}_t), \tilde{\mathbf{\Lambda}}_t^{-1}\nabla f(\mathbf{w}_t)\right\rangle. \tag{21}$$

*Proof.* It holds that

$$
\begin{aligned}
\left\langle \nabla f(\mathbf{w}_t), \left(\tilde{\mathbf{\Lambda}}_t^{-1} - \mathbf{\Lambda}_t^{-1}\right)\mathbf{g}_t\right\rangle &= \sum_{j=1}^d \nabla f_{t,j}\mathbf{g}_{t,j}\left(\frac{1}{\tilde{\lambda}_{t,j}} - \frac{1}{\lambda_{t,j}}\right) \\
&= \sum_{j=1}^d \frac{\nabla f_{t,j}\mathbf{g}_{t,j}\left(\mathbf{g}_{t,j}^2 - \mathbb{E}_t\left[\mathbf{g}_{t,j}^2\right]\right)}{\tilde{\lambda}_{t,j}\lambda_{t,j}(\lambda_{t,j} + \tilde{\lambda}_{t,j})} \\
&\le \sum_{j=1}^d \frac{|\nabla f_{t,j}||\mathbf{g}_{t,j}|\left|\mathbf{g}_{t,j}^2 - \mathbb{E}_t\left[\mathbf{g}_{t,j}^2\right]\right|}{\tilde{\lambda}_{t,j}\lambda_{t,j}(\lambda_{t,j} + \tilde{\lambda}_{t,j})} \\
&\le \sum_{j=1}^d \frac{|\nabla f_{t,j}||\mathbf{g}_{t,j}|\left||\mathbf{g}_{t,j}| - \sqrt{\mathbb{E}_t\left[\mathbf{g}_{t,j}^2\right]}\right|}{\tilde{\lambda}_{t,j}\lambda_{t,j}},
\end{aligned}
\tag{22}
$$

where in the first inequality we take the properties of absolute values. In the last inequality, we use the fact that

$$\tilde{\lambda}_{t,j}^2 = \lambda_{t-1,j}^2 + \mathbb{E}_t\left[\mathbf{g}_{t,j}^2\right] \ge \mathbb{E}_t\left[\mathbf{g}_{t,j}^2\right] \quad \text{and} \quad \lambda_{t,j}^2 = \lambda_{t-1,j}^2 + \mathbf{g}_{t,j}^2 \ge \mathbf{g}_{t,j}^2.$$

Then we consider an arbitrary coordinate $j \in [d]$. By applying inequality (31) with

$$c = \frac{\mathbb{E}_t\left[\left(|\mathbf{g}_{t,j}| - \sqrt{\mathbb{E}_t\left[\mathbf{g}_{t,j}^2\right]}\right)^2\right]}{\tilde{\lambda}_{t,j}}, \quad x = \frac{|\mathbf{g}_{t,j}|}{\lambda_{t,j}}, \quad y = \frac{\left||\mathbf{g}_{t,j}| - \sqrt{\mathbb{E}_t\left[\mathbf{g}_{t,j}^2\right]}\right||\nabla f_{t,j}|}{\tilde{\lambda}_{t,j}},$$

it holds that

$$\mathbb{E}_t\left[\frac{|\nabla f_{t,j}|\,|\mathbf{g}_{t,j}|\,\left|\,|\mathbf{g}_{t,j}| - \sqrt{\mathbb{E}_t\left[\mathbf{g}_{t,j}^2\right]}\,\right|}{\tilde{\lambda}_{t,j}\lambda_{t,j}}\right] \overset{(31)}{\leq} \frac{c}{2}\mathbb{E}_t\left[\frac{\mathbf{g}_{t,j}^2}{\lambda_{t,j}^2}\right] + \frac{1}{2c}\frac{\nabla f_{t,j}^2}{\tilde{\lambda}_{t,j}^2}\mathbb{E}_t\left[\left|\,|\mathbf{g}_{t,j}| - \sqrt{\mathbb{E}_t\left[\mathbf{g}_{t,j}^2\right]}\,\right|^2\right]$$

$$= \frac{\mathbb{E}_t\left[\left(|\mathbf{g}_{t,j}| - \sqrt{\mathbb{E}_t\left[\mathbf{g}_{t,j}^2\right]}\right)^2\right]}{2\tilde{\lambda}_{t,j}}\mathbb{E}_t\left[\frac{\mathbf{g}_{t,j}^2}{\lambda_{t,j}^2}\right] + \frac{\tilde{\lambda}_{t,j}}{2}\frac{\nabla f_{t,j}^2}{\tilde{\lambda}_{t,j}^2}$$

$$\leq \sqrt{\mathbb{E}_t\left[\left(|\mathbf{g}_{t,j}| - \sqrt{\mathbb{E}_t\left[\mathbf{g}_{t,j}^2\right]}\right)^2\right]}\mathbb{E}_t\left[\frac{\mathbf{g}_{t,j}^2}{\lambda_{t,j}^2}\right] + \frac{\nabla f_{t,j}^2}{2\tilde{\lambda}_{t,j}}$$

$$\overset{(5)}{\leq} 2\boldsymbol{\sigma}_j\mathbb{E}_t\left[\frac{\mathbf{g}_{t,j}^2}{\lambda_{t,j}^2}\right] + \frac{\nabla f_{t,j}^2}{2\tilde{\lambda}_{t,j}},$$

where in the second inequality we use the fact that

$$\mathbb{E}_t\left[\left(|\mathbf{g}_{t,j}| - \sqrt{\mathbb{E}_t\left[\mathbf{g}_{t,j}^2\right]}\right)^2\right] = 2\mathbb{E}_t\left[\mathbf{g}_{t,j}^2\right] - 2\mathbb{E}_t\left[|\mathbf{g}_{t,j}|\right]\sqrt{\mathbb{E}_t\left[\mathbf{g}_{t,j}^2\right]} \leq 2\mathbb{E}_t\left[\mathbf{g}_{t,j}^2\right]$$

$$\leq 2\tilde{\lambda}_{t,j}\sqrt{\mathbb{E}_t\left[\mathbf{g}_{t,j}^2\right]}$$

and the last inequality is based on the fact that

$$\mathbb{E}_t\left[\left(|\mathbf{g}_{t,j}| - \sqrt{\mathbb{E}_t\left[\mathbf{g}_{t,j}^2\right]}\right)^2\right] = 2\mathbb{E}_t\left[\mathbf{g}_{t,j}^2\right] - 2\mathbb{E}_t\left[|\mathbf{g}_{t,j}|\right]\sqrt{\mathbb{E}_t\left[\mathbf{g}_{t,j}^2\right]}$$

$$= 2\sqrt{\mathbb{E}_t\left[\mathbf{g}_{t,j}^2\right]}\left(\sqrt{\mathbb{E}_t\left[\mathbf{g}_{t,j}^2\right]} - \mathbb{E}_t\left[|\mathbf{g}_{t,j}|\right]\right)$$

$$= 2\sqrt{\mathbb{E}_t\left[\mathbf{g}_{t,j}^2\right]}\frac{\mathbb{E}_t\left[\mathbf{g}_{t,j}^2\right] - \mathbb{E}_t\left[|\mathbf{g}_{t,j}|\right]^2}{\sqrt{\mathbb{E}_t\left[\mathbf{g}_{t,j}^2\right]} + \mathbb{E}_t\left[|\mathbf{g}_{t,j}|\right]}$$

$$\leq 2\sqrt{\mathbb{E}_t\left[\mathbf{g}_{t,j}^2\right]}\frac{\mathbb{E}_t\left[\mathbf{g}_{t,j}^2\right] - \mathbb{E}_t\left[\mathbf{g}_{t,j}\right]^2}{\sqrt{\mathbb{E}_t\left[\mathbf{g}_{t,j}^2\right]} + \mathbb{E}_t\left[|\mathbf{g}_{t,j}|\right]}$$

$$\overset{(5)}{\leq} \frac{2\sqrt{\mathbb{E}_t\left[\mathbf{g}_{t,j}^2\right]}\boldsymbol{\sigma}_j^2}{\sqrt{\mathbb{E}_t\left[\mathbf{g}_{t,j}^2\right]} + \mathbb{E}_t\left[|\mathbf{g}_{t,j}|\right]} \leq 2\boldsymbol{\sigma}_j^2.$$

Thus by substituting into (22), we can obtain that

$$\mathbb{E}_t\left[\left\langle \nabla f(\mathbf{w}_t), \left(\tilde{\boldsymbol{\Lambda}}_t^{-1} - \boldsymbol{\Lambda}_t^{-1}\right)\mathbf{g}_t\right\rangle\right] \leq \mathbb{E}_t\left[\sum_{j=1}^d \frac{|\nabla f_{t,j}|\,|\mathbf{g}_{t,j}|\,\left|\,|\mathbf{g}_{t,j}| - \sqrt{\mathbb{E}_t\left[\mathbf{g}_{t,j}^2\right]}\,\right|}{\tilde{\lambda}_{t,j}\lambda_{t,j}}\right]$$

$$\leq \sum_{j=1}^d 2\boldsymbol{\sigma}_j\mathbb{E}_t\left[\frac{\mathbf{g}_{t,j}^2}{\lambda_{t,j}^2}\right] + \sum_{j=1}^d \frac{\nabla f_{t,j}^2}{2\tilde{\lambda}_{t,j}}$$

$$= \sum_{j=1}^d 2\boldsymbol{\sigma}_j\mathbb{E}_t\left[\frac{\mathbf{g}_{t,j}^2}{\lambda_{t,j}^2}\right] + \frac{1}{2}\left\langle \nabla f(\mathbf{w}_t), \tilde{\boldsymbol{\Lambda}}_t^{-1}\nabla f(\mathbf{w}_t)\right\rangle,$$

which concludes the proof. $\qquad\square$

**Lemma F.2** (Bound of $\mathbf{L}_{1,j}\,|\nabla f_{t,j}|\frac{\mathbf{g}_{t,j}^2}{\lambda_{t,j}^2}$). *Under the same settings as Theorem 5.3, it holds that*

$$\mathbb{E}_t\left[\sum_{j=1}^d \mathbf{L}_{1,j}\,|\nabla f_{t,j}|\frac{\mathbf{g}_{t,j}^2}{\lambda_{t,j}^2}\right] \leq 2\,\|\mathbf{L}_1\|_\infty\left\langle \nabla f(\mathbf{w}_t), \tilde{\boldsymbol{\Lambda}}_t^{-1}\nabla f(\mathbf{w}_t)\right\rangle + \sum_{j=1}^d \mathbf{L}_{1,j}\boldsymbol{\sigma}_j\mathbb{E}_t\left[\frac{\mathbf{g}_{t,j}^2}{\lambda_{t,j}^2}\right]. \quad (23)$$

*Proof.* Let us first consider an arbitrary coordinate $j \in [d]$ and two cases regarding $|\nabla f_{t,j}|$.

**(1)** If $|\nabla f_{t,j}| \leq \boldsymbol{\sigma}_j$, we have

$$\mathbb{E}_t \left[ \mathbf{L}_{1,j} \left| \nabla f_{t,j} \right| \frac{\mathbf{g}_{t,j}^2}{\lambda_{t,j}^2} \right] = \mathbf{L}_{1,j} \left| \nabla f_{t,j} \right| \mathbb{E}_t \left[ \frac{\mathbf{g}_{t,j}^2}{\lambda_{t,j}^2} \right] \leq \mathbf{L}_{1,j} \boldsymbol{\sigma}_j \mathbb{E}_t \left[ \frac{\mathbf{g}_{t,j}^2}{\lambda_{t,j}^2} \right].$$

**(2)** Else, we have $|\nabla f_{t,j}| \geq \boldsymbol{\sigma}_j$. In this case, we have

$$\mathbb{E}_t \left[ \frac{\mathbf{g}_{t,j}^2}{\lambda_{t,j}^2} \right] = \mathbb{E}_t \left[ \frac{\mathbf{g}_{t,j}^2}{\lambda_{t-1,j}^2 + \mathbf{g}_{t,j}^2} \right] \leq \frac{\mathbb{E}_t \left[ \mathbf{g}_{t,j}^2 \right]}{\lambda_{t-1,j}^2 + \mathbb{E}_t \left[ \mathbf{g}_{t,j}^2 \right]} \stackrel{(5)}{\leq} \frac{\nabla f_{t,j}^2 + \boldsymbol{\sigma}_j^2}{\lambda_{t-1,j}^2 + \mathbb{E}_t \left[ \mathbf{g}_{t,j}^2 \right]} \leq \frac{2 \nabla f_{t,j}^2}{\tilde{\lambda}_{t,j}^2},$$

where we use the Jensen inequality of convex function $h(x) = \frac{-x}{a^2 + x}$ in the first inequality. Therefore,

$$\mathbf{L}_{1,j} \left| \nabla f_{t,j} \right| \mathbb{E}_t \left[ \frac{\mathbf{g}_{t,j}^2}{\lambda_{t,j}^2} \right] \leq 2 \mathbf{L}_{1,j} \left| \nabla f_{t,j} \right| \frac{\nabla f_{t,j}^2}{\tilde{\lambda}_{t,j}^2} \leq 2 \mathbf{L}_{1,j} \frac{\nabla f_{t,j}^2}{\tilde{\lambda}_{t,j}},$$

where in the last inequality we use the fact that $|\nabla f_{t,j}| = |\mathbb{E}[\mathbf{g}_{t,j}]| \leq \sqrt{\mathbb{E}\left[\mathbf{g}_{t,j}^2\right]} \leq \tilde{\lambda}_{t,j}$. By combining the two cases, we have

$$\mathbf{L}_{1,j} \left| \nabla f_{t,j} \right| \mathbb{E}_t \left[ \frac{\mathbf{g}_{t,j}^2}{\lambda_{t,j}^2} \right] \leq 2 \mathbf{L}_{1,j} \frac{\nabla f_{t,j}^2}{\tilde{\lambda}_{t,j}} + \mathbf{L}_{1,j} \boldsymbol{\sigma}_j \mathbb{E}_t \left[ \frac{\mathbf{g}_{t,j}^2}{\lambda_{t,j}^2} \right]$$

and by summing up, we have

$$\mathbb{E}_t \left[ \sum_{j=1}^d \mathbf{L}_{1,j} \left| \nabla f_{t,j} \right| \frac{\mathbf{g}_{t,j}^2}{\lambda_{t,j}^2} \right] \leq 2 \left\| \mathbf{L}_1 \right\|_\infty \left\langle \nabla f(\mathbf{w}_t), \tilde{\mathbf{\Lambda}}_t^{-1} \nabla f(\mathbf{w}_t) \right\rangle + \sum_{j=1}^d \mathbf{L}_{1,j} \boldsymbol{\sigma}_j \mathbb{E}_t \left[ \frac{\mathbf{g}_{t,j}^2}{\lambda_{t,j}^2} \right].$$

$\square$

**Lemma F.3** (Bound of $\sum_{t=0}^{T-1} \frac{\mathbf{g}_{t,j}^2}{\lambda_{t,j}^2}$). *Under the same settings of Theorem 5.3, for all $j \in [d]$, it holds*

$$\mathbb{E} \left[ \sum_{t=0}^{T-1} \frac{\mathbf{g}_{t,j}^2}{\lambda_{t,j}^2} \right] \leq 2 \ln \left( \mathbb{E} \left[ \frac{\operatorname{tr} \left( \mathbf{\Lambda}_{T-1} \right)}{\epsilon} \right] \right). \tag{24}$$

*Proof.* It holds that

$$\sum_{t=0}^{T-1} \mathbb{E} \left[ \frac{\mathbf{g}_{t,j}^2}{\lambda_{t,j}^2} \right] \stackrel{(32)}{\leq} \mathbb{E} \left[ \ln \left( \frac{\lambda_{T-1,j}^2}{\epsilon^2} \right) \right] \leq \ln \left( 2 \mathbb{E} \left[ \frac{\lambda_{T-1,j}}{\epsilon} \right] \right) \leq 2 \ln \left( \mathbb{E} \left[ \frac{\operatorname{tr} \left( \mathbf{\Lambda}_{T-1} \right)}{\epsilon} \right] \right),$$

where the first inequality is based on Lemma G.3 and the fact that $\lambda_{t,j}^2 = \lambda_{t-1,j}^2 + \mathbf{g}_{t,j}^2$, and the second inequality is based on Jensen's inequality. $\square$

**Lemma F.4** (Bound of $\operatorname{tr}\left(\mathbf{\Lambda}_t\right)$). *Under the same settings of Theorem 5.3, it holds that*

$$\mathbb{E}\left[\operatorname{tr}\left(\mathbf{\Lambda}_{T-1}\right)\right] \leq 2 d\epsilon + \frac{4}{\eta}(f(\mathbf{w}_0) - f^*)$$
$$+ 5 \left( \eta \left\| \mathbf{L}_0 \right\|_1 + 3\sqrt{T} \left\| \boldsymbol{\sigma} \right\|_1 \right) \ln \left( \frac{2\eta \left\| \mathbf{L}_0 \right\|_1 + 5\sqrt{T} \left\| \boldsymbol{\sigma} \right\|_1}{\epsilon} + \mathrm{e} \right). \tag{25}$$

*Proof.* As we take $\eta \leq 1/\|\mathbf{L}_1\|_\infty$, we have $\|\mathbf{w}_{t+1} - \mathbf{w}_t\|_{\mathbf{L}_1} \leq \sqrt{d}$. Then from Assumption 3.3 and Lemma C.3 it holds that

$$f(\mathbf{w}_{t+1}) \overset{(15)}{\leq} f(\mathbf{w}_t) + \langle \nabla f(\mathbf{w}_t), \mathbf{w}_{t+1} - \mathbf{w}_t \rangle + \frac{1}{2} \sum_{j=1}^d (\mathbf{L}_{0,j} + \mathbf{L}_{1,j} |\nabla f_{t,j}|) |\mathbf{w}_{t+1} - \mathbf{w}_t|_j^2$$

$$= f(\mathbf{w}_t) - \eta \langle \nabla f(\mathbf{w}_t), \mathbf{\Lambda}_t^{-1} \mathbf{g}_t \rangle + \frac{\eta^2}{2} \sum_{j=1}^d (\mathbf{L}_{0,j} + \mathbf{L}_{1,j} |\nabla f_{t,j}|) \frac{\mathbf{g}_{t,j}^2}{\lambda_{t,j}^2}$$

$$= f(\mathbf{w}_t) - \eta \langle \mathbf{g}_t, \mathbf{\Lambda}_t^{-1} \mathbf{g}_t \rangle + \frac{\eta^2}{2} \sum_{j=1}^d (\mathbf{L}_{0,j} + \mathbf{L}_{1,j} |\nabla f_{t,j}|) \frac{\mathbf{g}_{t,j}^2}{\lambda_{t,j}^2} + \eta \langle \mathbf{n}_t, \mathbf{\Lambda}_t^{-1} \mathbf{g}_t \rangle$$

$$\leq f(\mathbf{w}_t) - \eta \langle \mathbf{g}_t, \mathbf{\Lambda}_t^{-1} \mathbf{g}_t \rangle + \frac{\eta^2}{2} \sum_{j=1}^d (\mathbf{L}_{0,j} + \mathbf{L}_{1,j}(|\mathbf{g}_{t,j}| + |\mathbf{n}_{t,j}|)) \frac{\mathbf{g}_{t,j}^2}{\lambda_{t,j}^2} + \eta \langle \mathbf{n}_t, \mathbf{\Lambda}_t^{-1} \mathbf{g}_t \rangle$$

$$= f(\mathbf{w}_t) - \eta \langle \mathbf{g}_t, \mathbf{\Lambda}_t^{-1} \mathbf{g}_t \rangle + \frac{\eta^2}{2} \sum_{j=1}^d \mathbf{L}_{0,j} \frac{\mathbf{g}_{t,j}^2}{\lambda_{t,j}^2} + \frac{\eta^2}{2} \sum_{j=1}^d \mathbf{L}_{1,j} \frac{|\mathbf{g}_{t,j}|^3}{\lambda_{t,j}^2}$$

$$+ \frac{\eta^2}{2} \sum_{j=1}^d \mathbf{L}_{1,j} |\mathbf{n}_{t,j}| \frac{\mathbf{g}_{t,j}^2}{\lambda_{t,j}^2} + \eta \langle \mathbf{n}_t, \mathbf{\Lambda}_t^{-1} \mathbf{g}_t \rangle$$

$$\leq f(\mathbf{w}_t) - \eta \langle \mathbf{g}_t, \mathbf{\Lambda}_t^{-1} \mathbf{g}_t \rangle + \frac{\eta^2}{2} \sum_{j=1}^d \mathbf{L}_{0,j} \frac{\mathbf{g}_{t,j}^2}{\lambda_{t,j}^2} + \frac{\eta^2}{2} \sum_{j=1}^d \mathbf{L}_{1,j} \frac{|\mathbf{g}_{t,j}|^3}{\lambda_{t,j}^2}$$

$$+ \frac{\eta^2}{2} \sum_{j=1}^d \mathbf{L}_{1,j} |\mathbf{n}_{t,j}| \frac{\mathbf{g}_{t,j}^2}{\lambda_{t,j}^2} + \eta \sum_{j=1}^d |\mathbf{n}_{t,j}| \frac{|\mathbf{g}_{t,j}|}{\lambda_{t,j}},$$

where we use absolute value inequality. Then we deal with the terms separately by

$$\frac{\eta^2}{2} \sum_{j=1}^d \mathbf{L}_{1,j} \frac{|\mathbf{g}_{t,j}|^3}{\lambda_{t,j}^2} \leq \frac{\eta^2}{2} \|\mathbf{L}_1\|_\infty \sum_{j=1}^d \frac{|\mathbf{g}_{t,j}|^3}{\lambda_{t,j}^2} \leq \frac{\eta^2}{2} \|\mathbf{L}_1\|_\infty \sum_{j=1}^d \frac{\mathbf{g}_{t,j}^2}{\lambda_{t,j}} = \frac{\eta^2}{2} \|\mathbf{L}_1\|_\infty \langle \mathbf{g}_t, \mathbf{\Lambda}_t^{-1} \mathbf{g}_t \rangle$$

$$\leq \frac{\eta}{2} \langle \mathbf{g}_t, \mathbf{\Lambda}_t^{-1} \mathbf{g}_t \rangle,$$

where we use the fact that $|\mathbf{g}_{t,j}| \leq \lambda_{t,j}$ and $\eta \leq 1/\|\mathbf{L}_1\|_\infty$. Similarly, we can also obtain that

$$\frac{\eta^2}{2} \sum_{j=1}^d \mathbf{L}_{1,j} |\mathbf{n}_{t,j}| \frac{\mathbf{g}_{t,j}^2}{\lambda_{t,j}^2} \leq \frac{\eta^2}{2} \sum_{j=1}^d \mathbf{L}_{1,j} |\mathbf{n}_{t,j}| \frac{|\mathbf{g}_{t,j}|}{\lambda_{t,j}} \leq \frac{\eta}{2} \sum_{j=1}^d |\mathbf{n}_{t,j}| \frac{|\mathbf{g}_{t,j}|}{\lambda_{t,j}},$$

where we use the fact that $|\mathbf{g}_{t,j}| \leq \lambda_{t,j}$ and $\eta \leq 1/\|\mathbf{L}_1\|_\infty$. By combining the bounds of the two terms, we can obtain that

$$f(\mathbf{w}_{t+1}) \leq f(\mathbf{w}_t) - \frac{\eta}{2} \langle \mathbf{g}_t, \mathbf{\Lambda}_t^{-1} \mathbf{g}_t \rangle + \frac{\eta^2}{2} \sum_{j=1}^d \mathbf{L}_{0,j} \frac{\mathbf{g}_{t,j}^2}{\lambda_{t,j}^2} + \frac{3\eta}{2} \sum_{j=1}^d |\mathbf{n}_{t,j}| \frac{|\mathbf{g}_{t,j}|}{\lambda_{t,j}}.$$

Then after summation in $t$ and rearrangement, it holds that

$$\sum_{t=0}^{T-1} \frac{\eta}{2} \langle \mathbf{g}_t, \mathbf{\Lambda}_t^{-1} \mathbf{g}_t \rangle \leq \sum_{t=0}^{T-1} [f(\mathbf{w}_t) - f(\mathbf{w}_{t+1})] + \frac{\eta^2}{2} \sum_{j=1}^d \mathbf{L}_{0,j} \sum_{t=0}^{T-1} \frac{\mathbf{g}_{t,j}^2}{\lambda_{t,j}^2} + \frac{3\eta}{2} \sum_{j=1}^d \sum_{t=0}^{T-1} |\mathbf{n}_{t,j}| \frac{|\mathbf{g}_{t,j}|}{\lambda_{t,j}}$$

$$\leq f(\mathbf{w}_0) - f(\mathbf{w}_T) + \frac{\eta^2}{2} \sum_{j=1}^d \mathbf{L}_{0,j} \sum_{t=0}^{T-1} \frac{\mathbf{g}_{t,j}^2}{\lambda_{t,j}^2} + \frac{3\eta}{2} \sum_{j=1}^d \sqrt{\sum_{t=0}^{T-1} \mathbf{n}_{t,j}^2} \sqrt{\sum_{t=0}^{T-1} \frac{\mathbf{g}_{t,j}^2}{\lambda_{t,j}^2}},$$

where in the last inequality we use Cauchy-Schwarz Inequality. Moreover, it holds that

$$\sum_{t=0}^{T-1}\left\langle \mathbf{g}_t, \mathbf{\Lambda}_t^{-1}\mathbf{g}_t\right\rangle = \sum_{j=1}^{d}\sum_{t=0}^{T-1}\frac{\mathbf{g}_{t,j}^2}{\lambda_{t,j}} \geq \sum_{j=1}^{d}\sum_{t=0}^{T-1}\frac{\mathbf{g}_{t,j}^2}{\lambda_{t,j}+\lambda_{t-1,j}} = \sum_{t=0}^{T-1}\frac{\lambda_{t,j}^2-\lambda_{t-1,j}^2}{\lambda_{t,j}+\lambda_{t-1,j}}$$

$$= \sum_{j=1}^{d}\sum_{t=0}^{T-1}(\lambda_{t,j}-\lambda_{t-1,j}) = \operatorname{tr}\left(\mathbf{\Lambda}_{T-1}-\mathbf{\Lambda}_{-1}\right),$$

where $\mathbf{\Lambda}_{-1} = \epsilon\mathbf{I}_d$. Then by combining the two inequalities together with Lemma F.3 and taking expectation, we can obtain that

$$\mathbb{E}\left[\operatorname{tr}\left(\mathbf{\Lambda}_{T-1}\right)\right] \leq \operatorname{tr}\left(\mathbf{\Lambda}_{-1}\right) + \frac{2}{\eta}\mathbb{E}[f(\mathbf{w}_0)-f(\mathbf{w}_T)] + 2\eta\sum_{j=1}^{d}\mathbf{L}_{0,j}\mathbb{E}\left[\sum_{t=0}^{T-1}\frac{\mathbf{g}_{t,j}^2}{\lambda_{t,j}^2}\right]$$

$$+ 3\sum_{j=1}^{d}\mathbb{E}\left[\sqrt{\sum_{t=0}^{T-1}\mathbf{n}_{t,j}^2}\cdot\sqrt{\sum_{t=0}^{T-1}\frac{\mathbf{g}_{t,j}^2}{\lambda_{t,j}^2}}\right]$$

$$\overset{(24)}{\leq} d\epsilon + \frac{2}{\eta}\mathbb{E}[f(\mathbf{w}_0)-f(\mathbf{w}_T)] + 2\eta\sum_{j=1}^{d}\mathbf{L}_{0,j}\ln\left(\mathbb{E}\left[\frac{\operatorname{tr}\left(\mathbf{\Lambda}_{T-1}\right)}{\epsilon}\right]\right)$$

$$+ 3\sum_{j=1}^{d}\mathbb{E}\left[\sqrt{\sum_{t=0}^{T-1}\mathbf{n}_{t,j}^2}\cdot\sqrt{\sum_{t=0}^{T-1}\frac{\mathbf{g}_{t,j}^2}{\lambda_{t,j}^2}}\right]$$

$$\leq d\epsilon + \frac{2}{\eta}\mathbb{E}[f(\mathbf{w}_0)-f(\mathbf{w}_T)] + 2\eta\sum_{j=1}^{d}\mathbf{L}_{0,j}\ln\left(\mathbb{E}\left[\frac{\operatorname{tr}\left(\mathbf{\Lambda}_{T-1}\right)}{\epsilon}\right]\right)$$

$$+ 3\sum_{j=1}^{d}\sqrt{\mathbb{E}\left[\sum_{t=0}^{T-1}\mathbf{n}_{t,j}^2\right]}\cdot\sqrt{\mathbb{E}\left[\sum_{t=0}^{T-1}\frac{\mathbf{g}_{t,j}^2}{\lambda_{t,j}^2}\right]}$$

$$\overset{(5),(24)}{\leq} d\epsilon + \frac{2}{\eta}\mathbb{E}[f(\mathbf{w}_0)-f(\mathbf{w}_T)] + 2\eta\sum_{j=1}^{d}\mathbf{L}_{0,j}\ln\left(\mathbb{E}\left[\frac{\operatorname{tr}\left(\mathbf{\Lambda}_{T-1}\right)}{\epsilon}\right]\right)$$

$$+ 3\sqrt{T}\left\|\boldsymbol{\sigma}\right\|_1\cdot\sqrt{2\mathbb{E}\left[\ln\left(\frac{\operatorname{tr}\left(\mathbf{\Lambda}_{T-1}\right)}{\epsilon}\right)\right]}$$

$$\leq d\epsilon + \frac{2}{\eta}\mathbb{E}[f(\mathbf{w}_0)-f(\mathbf{w}_T)] + \left(2\eta\left\|\mathbf{L}_0\right\|_1 + 5\sqrt{T}\left\|\boldsymbol{\sigma}\right\|_1\right)\ln\left(\frac{\mathbb{E}\left[\operatorname{tr}\left(\mathbf{\Lambda}_{T-1}\right)^2\right]}{\epsilon}\right),$$

where the second inequality is based on the Jensen inequality and the third inequality is based on the Cauchy-Schwarz inequality. Then by taking

$$x = \frac{\mathbb{E}[\operatorname{tr}\left(\mathbf{\Lambda}_{T-1}\right)]}{\epsilon}, \quad C_1 = \frac{2\eta\left\|\mathbf{L}_0\right\|_1 + 5\sqrt{T}\left\|\boldsymbol{\sigma}\right\|_1}{\epsilon}, \quad \text{and} \quad C_0 = \frac{d\epsilon + \frac{1}{\eta}[f(\mathbf{w}_0)-f(\mathbf{w}_T)]}{\epsilon}$$

in Lemma G.4, and using Assumption 3.2, we can obtain that

$$\mathbb{E}\left[\operatorname{tr}\left(\mathbf{\Lambda}_{T-1}\right)\right] \leq 2d\epsilon + \frac{4}{\eta}(f(\mathbf{w}_0)-f^*)$$

$$+ 5\left(2\eta\left\|\mathbf{L}_0\right\|_1 + 5\sqrt{T}\left\|\boldsymbol{\sigma}\right\|_1\right)\ln\left(\frac{2\eta\left\|\mathbf{L}_0\right\|_1 + 5\sqrt{T}\left\|\boldsymbol{\sigma}\right\|_1}{\epsilon} + \mathrm{e}\right),$$

which concludes the proof. $\qquad\square$

Then we are ready to prove Theorem 5.3.

*Proof of Theorem 5.3.* As we take $\eta \leq 1/\|\mathbf{L}_1\|_\infty$, we have $\|\mathbf{w}_{t+1} - \mathbf{w}_t\|_{\mathbf{L}_1} \leq \sqrt{d}$. Then from Assumption 3.3 and Lemma C.3 it holds that

$$f(\mathbf{w}_{t+1}) \overset{(15)}{\leq} f(\mathbf{w}_t) + \langle \nabla f(\mathbf{w}_t), \mathbf{w}_{t+1} - \mathbf{w}_t \rangle + \frac{1}{2} \sum_{j=1}^d \left( \mathbf{L}_{0,j} + \mathbf{L}_{1,j} |\nabla f_{t,j}| \right) |\mathbf{w}_{t+1} - \mathbf{w}_t|_j^2$$

$$= f(\mathbf{w}_t) - \eta \langle \nabla f(\mathbf{w}_t), \mathbf{\Lambda}_t^{-1} \mathbf{g}_t \rangle + \frac{\eta^2}{2} \sum_{j=1}^d \left( \mathbf{L}_{0,j} + \mathbf{L}_{1,j} |\nabla f_{t,j}| \right) \frac{\mathbf{g}_{t,j}^2}{\lambda_{t,j}^2}.$$

Then by taking expectation and summation on $t$ we can obtain that

$$\sum_{t=0}^{T-1} \mathbb{E}_t[f(\mathbf{w}_{t+1})] \leq \sum_{t=0}^{T-1} f(\mathbf{w}_t) - \eta \sum_{t=0}^{T-1} \mathbb{E}_t \left[ \langle \nabla f(\mathbf{w}_t), \mathbf{\Lambda}_t^{-1} \mathbf{g}_t \rangle \right] + \frac{\eta^2}{2} \sum_{j=1}^d \mathbf{L}_{0,j} \sum_{t=0}^{T-1} \mathbb{E}_t \left[ \frac{\mathbf{g}_{t,j}^2}{\lambda_{t,j}^2} \right]$$

$$+ \frac{\eta^2}{2} \sum_{j=1}^d \mathbf{L}_{1,j} \sum_{t=0}^{T-1} |\nabla f_{t,j}| \, \mathbb{E}_t \left[ \frac{\mathbf{g}_{t,j}^2}{\lambda_{t,j}^2} \right]$$

$$\overset{(23)}{\leq} \sum_{t=0}^{T-1} f(\mathbf{w}_t) - \eta \sum_{t=0}^{T-1} \mathbb{E}_t \left[ \langle \nabla f(\mathbf{w}_t), \mathbf{\Lambda}_t^{-1} \mathbf{g}_t \rangle \right] + \frac{\eta^2}{2} \sum_{j=1}^d \mathbf{L}_{0,j} \sum_{t=0}^{T-1} \mathbb{E}_t \left[ \frac{\mathbf{g}_{t,j}^2}{\lambda_{t,j}^2} \right]$$

$$+ \eta^2 \|\mathbf{L}_1\|_\infty \left( \sum_{t=0}^{T-1} \langle \nabla f(\mathbf{w}_t), \tilde{\mathbf{\Lambda}}_t^{-1} \nabla f(\mathbf{w}_t) \rangle + \frac{1}{2} \sum_{j=1}^d \boldsymbol{\sigma}_j \sum_{t=0}^{T-1} \mathbb{E}_t \left[ \frac{\mathbf{g}_{t,j}^2}{\lambda_{t,j}^2} \right] \right). \tag{26}$$

We first consider the second term on the right hand side of (26). It holds that

$$- \sum_{t=0}^{T-1} \mathbb{E} \left[ \langle \nabla f(\mathbf{w}_t), \mathbf{\Lambda}_t^{-1} \mathbf{g}_t \rangle \right] = - \sum_{t=0}^{T-1} \mathbb{E} \left[ \langle \nabla f(\mathbf{w}_t), \tilde{\mathbf{\Lambda}}_t^{-1} \mathbf{g}_t \rangle \right]$$

$$+ \sum_{t=0}^{T-1} \mathbb{E} \left[ \langle \nabla f(\mathbf{w}_t), \left( \tilde{\mathbf{\Lambda}}_t^{-1} - \mathbf{\Lambda}_t^{-1} \right) \mathbf{g}_t \rangle \right]$$

$$\overset{(21)}{\leq} - \frac{1}{2} \sum_{t=0}^{T-1} \mathbb{E} \left[ \langle \nabla f(\mathbf{w}_t), \tilde{\mathbf{\Lambda}}_t^{-1} \mathbf{g}_t \rangle \right] + \sum_{t=0}^{T-1} \sum_{j=1}^d 2 \boldsymbol{\sigma}_j \mathbb{E} \left[ \frac{\mathbf{g}_{t,j}^2}{\lambda_{t,j}^2} \right]$$

Therefore, after substituting the inequality and (24) into (26) and setting $\eta \leq \frac{1}{4\|\mathbf{L}_1\|_\infty}$, we have

$$\sum_{t=0}^{T-1} \mathbb{E} \left[ \langle \nabla f(\mathbf{w}_t), \tilde{\mathbf{\Lambda}}_t^{-1} \mathbf{g}_t \rangle \right] \leq \frac{4}{\eta} \mathbb{E} \left[ f(\mathbf{w}_0) - f(\mathbf{w}_T) \right] + (12 \|\boldsymbol{\sigma}\|_1 + 2 \|\mathbf{L}_0\|_1) \ln \left( \mathbb{E} \left[ \frac{(\mathrm{tr} \, (\mathbf{\Lambda}_{T-1})^2)}{\epsilon} \right] \right). \tag{27}$$

Moreover, for the left hand side of (27), we have

$$\mathbb{E} \left[ \langle \nabla f(\mathbf{w}_t), \tilde{\mathbf{\Lambda}}_t^{-1} \mathbf{g}_t \rangle \right] = \mathbb{E} \left[ \langle \nabla f(\mathbf{w}_t), \tilde{\mathbf{\Lambda}}_t^{-1} \nabla f(\mathbf{w}_t) \rangle \right] = \sum_{j=1}^d \mathbb{E} \left[ \frac{|\nabla f_{t,j}|^2}{\sqrt{\lambda_{t-1,j}^2 + \mathbb{E}_t [\mathbf{g}_{t,j}]^2}} \right]$$

$$\overset{(5)}{\geq} \sum_{j=1}^d \mathbb{E} \left[ \frac{|\nabla f_{t,j}|^2}{\sqrt{\lambda_{t-1,j}^2 + \nabla f_{t,j}^2 + \boldsymbol{\sigma}_j^2}} \right]$$

$$\geq \sum_{j=1}^d \mathbb{E} \left[ \frac{|\nabla f_{t,j}|^2}{\sqrt{\lambda_{T-1,j}^2 + \sum_{s=0}^{T-1} \nabla f_{s,j}^2 + \boldsymbol{\sigma}_j^2}} \right].$$

Thus by taking summation we have

$$
\begin{aligned}
\sum_{t=0}^{T-1} \mathbb{E}\left[\left\langle \nabla f(\mathbf{w}_t), \tilde{\mathbf{\Lambda}}_t^{-1}\mathbf{g}_t \right\rangle\right] &\geq \sum_{j=1}^{d}\sum_{t=0}^{T-1} \mathbb{E}\left[\frac{|\nabla f_{t,j}|^2}{\sqrt{\lambda_{T-1,j}^2 + \sum_{s=0}^{T-1}\nabla f_{s,j}^2 + \boldsymbol{\sigma}_j^2}}\right] \\
&= \sum_{j=1}^{d} \mathbb{E}\left[\frac{\sum_{t=0}^{T-1}|\nabla f_{t,j}|^2}{\sqrt{\lambda_{T-1,j}^2 + \sum_{s=0}^{T-1}\nabla f_{s,j}^2 + \boldsymbol{\sigma}_j^2}}\right] \\
&\geq \sum_{j=1}^{d} \frac{\mathbb{E}\left[\sqrt{\sum_{t=0}^{T-1}|\nabla f_{t,j}|^2}\right]^2}{\mathbb{E}\left[\sqrt{\lambda_{T-1,j}^2 + \sum_{s=0}^{T-1}\nabla f_{s,j}^2 + \boldsymbol{\sigma}_j^2}\right]} \\
&\geq \frac{\mathbb{E}\left[\sum_{j=1}^{d}\sqrt{\sum_{t=0}^{T-1}|\nabla f_{t,j}|^2}\right]^2}{\sum_{j=1}^{d}\mathbb{E}\left[\sqrt{\lambda_{T-1,j}^2 + \sum_{s=0}^{T-1}\nabla f_{s,j}^2 + \boldsymbol{\sigma}_j^2}\right]},
\end{aligned}
\tag{28}
$$

where in the second last and last inequality we apply Cauchy-Schwarz inequality

$$
\mathbb{E}[|XY|] \leq \mathbb{E}\left[|X|^2\right]^{\frac{1}{2}} \cdot \mathbb{E}\left[|Y|^2\right]^{\frac{1}{2}} \quad \text{and} \quad \sum_{j=1}^{d}|X_j Y_j| \leq \left(\sum_{j=1}^{d}X_j^2\right)^{\frac{1}{2}}\left(\sum_{j=1}^{d}Y_j^2\right)^{\frac{1}{2}}.
$$

We can further deal with the denominator such that for all $j \in [d]$,

$$
\begin{aligned}
\mathbb{E}\left[\sqrt{\lambda_{T-1,j}^2 + \sum_{s=0}^{T-1}\nabla f_{s,j}^2 + \boldsymbol{\sigma}_j^2}\right] &\leq \mathbb{E}\left[\sqrt{\sum_{s=0}^{T-1}\mathbf{g}_{s,j}^2}\right] + \mathbb{E}\left[\sqrt{\sum_{s=0}^{T-1}\nabla f_{s,j}^2 + \boldsymbol{\sigma}_j^2}\right] \\
&= \mathbb{E}\left[\sqrt{\sum_{s=0}^{T-1}\mathbf{g}_{s,j}^2}\right] + \mathbb{E}\left[\sqrt{\sum_{s=0}^{T-1}(\mathbf{g}_{s,j} - \mathbf{n}_{s,j})^2 + \boldsymbol{\sigma}_j^2}\right] \\
&\leq \mathbb{E}\left[\sqrt{\sum_{s=0}^{T-1}\mathbf{g}_{s,j}^2}\right] + \mathbb{E}\left[\sqrt{\sum_{s=0}^{T-1}2\mathbf{g}_{s,j}^2 + 2\mathbf{n}_{s,j}^2 + \boldsymbol{\sigma}_j^2}\right] \\
&\leq 3\mathbb{E}\left[\sqrt{\sum_{s=0}^{T-1}\mathbf{g}_{s,j}^2}\right] + \mathbb{E}\left[\sqrt{\sum_{s=0}^{T-1}2\mathbf{n}_{s,j}^2 + \boldsymbol{\sigma}_j^2}\right] \\
&\overset{(5)}{\leq} 3\mathbb{E}\left[\sqrt{\sum_{s=0}^{T-1}\mathbf{g}_{s,j}^2}\right] + \sqrt{T+1}\boldsymbol{\sigma}_j = 3\mathbb{E}\left[\lambda_{t,j}\right] + \sqrt{T+1}\boldsymbol{\sigma}_j
\end{aligned}
\tag{29}
$$

where the first and the third inequality holds as $\sqrt{a+b} \leq \sqrt{a} + \sqrt{b}$ for $a, b > 0$ and the second inequality holds as $(a+b)^2 \leq 2a^2 + 2b^2$ for $a, b \in \mathbb{R}$. Therefore, with Lemma F.1 bounding $\operatorname{tr}(\mathbf{\Lambda}_t)$, combining (27) with (28) and (29) we can obtain that

$$
\mathbb{E}\left[\sum_{j=1}^{d}\sqrt{\sum_{t=0}^{T-1}|\nabla f_{t,j}|^2}\right]^2 \overset{(27),(28)}{\leq} \sum_{j=1}^{d}\mathbb{E}\left[\sqrt{\lambda_{T-1,j}^2 + \sum_{s=0}^{T-1}\nabla f_{s,j}^2 + \boldsymbol{\sigma}_j}\right]
$$

$$\cdot \left( \frac{4}{\eta} \mathbb{E}[f(\mathbf{w}_0) - f(\mathbf{w}_T)] + (12 \|\boldsymbol{\sigma}\|_1 + 2\eta \|\mathbf{L}_0\|_1) \ln \left( \mathbb{E} \left[ \frac{(\operatorname{tr}(\mathbf{\Lambda}_{T-1})^2)}{\epsilon} \right] \right) \right)$$

$$\overset{(29)}{\leq} \left( 3\mathbb{E} \left[ \operatorname{tr}(\mathbf{\Lambda}_{T-1}) \right] + \sqrt{T+1} \|\boldsymbol{\sigma}\|_1 \right)$$
$$\cdot \left( \frac{4}{\eta} \mathbb{E}[f(\mathbf{w}_0) - f(\mathbf{w}_T)] + (12 \|\boldsymbol{\sigma}\|_1 + 2\eta \|\mathbf{L}_0\|_1) \ln \left( \mathbb{E} \left[ \frac{(\operatorname{tr}(\mathbf{\Lambda}_{T-1})^2)}{\epsilon} \right] \right) \right)$$

$$\overset{(25)}{\leq} \left( 30B(\eta) + 6d\epsilon + 76\sqrt{T+1}V \right) \cdot (4B(\eta) + 12V),$$

where we denote

$$B(\eta) = \frac{1}{\eta}(f(\mathbf{w}_0) - f^*) + \eta \|\mathbf{L}_0\|_1 C_{\log}, \quad V = \|\boldsymbol{\sigma}\|_1 C_{\log}$$

and

$$C_{\log} = \ln \left( 2d\epsilon + \frac{4}{\eta}(f(\mathbf{w}_0) - f^*) \right.$$
$$\left. + 5 \left( 2\eta \|\mathbf{L}_0\|_1 + 5\sqrt{T} \|\boldsymbol{\sigma}\|_1 \right) \ln \left( \frac{2\eta \|\mathbf{L}_0\|_1 + 5\sqrt{T} \|\boldsymbol{\sigma}\|_1}{\epsilon} + \mathrm{e} \right) \right)$$
$$= \mathcal{O}(\log T)$$

based on Lemma F.4. By taking $\eta = \min \left\{ \frac{1}{4\|\mathbf{L}_1\|_\infty}, \sqrt{\frac{\|\mathbf{L}_0\|_1}{\Delta}} \right\}$, where $\Delta \triangleq f(\mathbf{w}_0) - f^*$, we can obtain that

$$B(\eta) = \tilde{\mathcal{O}} \left( \sqrt{\|\mathbf{L}_0\|_1 \Delta} + \|\mathbf{L}_1\|_\infty \Delta \right).$$

Then combining the fact proven in Lemma G.5 that

$$\sum_{j=1}^d \sqrt{\sum_{t=0}^{T-1} |\nabla f_{t,j}|^2} \geq \sqrt{\sum_{t=0}^{T-1} \|\nabla f(\mathbf{w}_t)\|_1^2}$$

and dividing $T$ in both side, we can obtain that

$$\frac{1}{T} \left( \mathbb{E} \left[ \sqrt{\sum_{t=0}^{t-1} \|\nabla f(\mathbf{w}_t)\|_1^2} \right] \right)^2 = \tilde{\mathcal{O}} \left( \frac{\sqrt{\|\mathbf{L}_0\|_1 \Delta} \|\boldsymbol{\sigma}\|_1}{\sqrt{MT}} + \frac{\|\boldsymbol{\sigma}\|_1^2}{M\sqrt{T}} + \frac{\|\mathbf{L}_0\|_1 \Delta}{T} \right)$$
$$+ \tilde{\mathcal{O}} \left( \frac{\|\mathbf{L}_1\|_\infty \Delta \|\boldsymbol{\sigma}\|_1}{\sqrt{MT}} + \frac{\|\mathbf{L}_1\|_\infty^2 \Delta^2}{T} \right)$$
$$+ \tilde{\mathcal{O}} \left( \frac{d\epsilon \left( \sqrt{\|\mathbf{L}_0\|_1 \Delta} + \|\mathbf{L}_1\|_\infty \Delta \right)}{T} + \frac{d\epsilon \|\boldsymbol{\sigma}\|_1}{\sqrt{MT}} \right),$$

which concludes the proof. $\qquad \square$

## G   USEFUL LEMMAS

**Lemma G.1** (A useful inequality). *Assume a non-negative sequence $\{x_j\}_{j=1}^n$ and a positive sequence $\{s_j\}_{j=1}^n$ with $S = \sum_{i=1}^n s_j$, it holds that*

$$\frac{1}{S} \sum_{j=1}^n x_j \leq \sqrt{\frac{1}{S} \sum_{j=1}^n \frac{x_j^2}{s_j}}. \tag{30}$$

*The inequality holds as an equality if and only if for all $i = 1, \cdots, n$ and $j = 1, \cdots, n$,*
$$\frac{x_i}{s_i} = \frac{x_j}{s_j}.$$

*Proof.* We first multiply $S$ by both sides and take the square, then the right-hand side minus the left-hand side will be

$$\sum_{j=1}^{n} \frac{S}{s_j} x_j^2 - \left( \sum_{j=1}^{n} x_j \right)^2 = \sum_{j=1}^{n} \sum_{i=1}^{n} \frac{s_i}{s_j} x_j^2 - \sum_{j=1}^{n} \sum_{i=1}^{n} x_i x_j$$

$$= \sum_{j \neq i} \frac{s_i}{s_j} x_j^2 - \sum_{j \neq i} x_i x_j$$

$$= \sum_{j \neq i} \left( \sqrt{\frac{s_i}{s_j}} x_j - \sqrt{\frac{s_j}{s_i}} x_i \right)^2 \geq 0,$$

which concludes the proof. Note that this result is an application of the Cauchy-Schwarz inequality. $\square$

**Lemma G.2** (A basic inequality). *For $c \geq 0$ and $x, y \in \mathbb{R}$, it holds that*

$$|xy| \leq \frac{c}{2} x^2 + \frac{1}{2c} y^2. \tag{31}$$

**Lemma G.3** (sum of ratios with the denominator being the sum of past numerators). *Assume a non-negative sequence $(a_n)$ and $\epsilon > 0$. We define $b_n = \sum_{i=1}^{n} a_i$. Then it holds that*

$$\sum_{t=1}^{N} \frac{a_t}{b_t + \epsilon} \leq \ln \left( \frac{b_N + \epsilon}{\epsilon} \right). \tag{32}$$

*Proof.* It holds that

$$\frac{a_t}{b_t + \epsilon} \leq -\ln \left( 1 - \frac{a_t}{\epsilon + b_t} \right)$$

$$= \ln(b_t + \epsilon) - \ln(b_t - a_t + \epsilon)$$

$$= \ln(b_t + \epsilon) - \ln(b_{t-1} + \epsilon),$$

where the inequality is based on the fact that $x \leq \ln(1 + x)$ for all $x > -1$ and we set $b_0 = 0$. Then by summing up for 1 to $N$ we finish the proof. $\square$

**Lemma G.4** (Solving inequality $x \leq C_0 + C_1 \ln x$). *Assume $C_1 \geq C_0 \geq 0$ and $x > 0$. If $x \leq C_0 + C_1 \ln x$ (where $\ln$ denotes the natural logarithm), it holds that for $\zeta \geq 5$,*

$$x \leq 2C_0 + \zeta C_1 \ln(C_1 + e). \tag{33}$$

*Proof.* Denote $g(x) = x - C_1 \ln x - C_0$. Then we have $g$ is a convex function and attains uniform lower bound at $C_1$. For $x \geq C_1$, $g$ is a monotonically increasing function. Therefore, to verify (33), it is equivalent to verify that if $x = 2C_0 + \zeta C_1 \ln(C_1 + e)$, where $\zeta = 5$, then $x \geq C_0 + C_1 \ln x$. We begin the verification then.

Denote $z = 2C_0 + \zeta C_1 \ln(C_1 + e)$. Then we have

$$z - (C_0 + C_1 \ln z) = z - (C_0 + C_1 \ln (2C_0 + \zeta C_1 \ln(C_1 + e)))$$

$$\geq z - \left( C_0 + \frac{1}{2} \cdot 2C_0 + C_1 \ln (\zeta C_1 \ln(C_1 + e)) \right)$$

$$= C_1 \left( \zeta \ln(C_1 + e) - \ln (\zeta C_1 \ln(C_1 + e)) \right),$$

where the second inequality is based on the fact that

$$C_1 \ln (2C_0 + \zeta C_1 \ln(C_1 + e)) \leq \frac{1}{2} C_0 + C_1 \ln (\zeta C_1 \ln(C_1 + e))$$

for $C_0 \geq 0$ and $\zeta \geq 2$. Then let us consider function $h(y) = \zeta \ln(y + e) - \ln (\zeta y \ln(y + e))$ for $y \geq 0$ and we have $h(1) \geq 0$. We also have

$$h'(y) = \frac{\zeta}{y + e} - \frac{1}{\zeta y \ln(y + e)} \left( \zeta \ln(y + e) + \frac{\zeta y}{y + e} \right) = \frac{\zeta}{y + e} - \frac{1}{y} - \frac{1}{(y + e) \ln(y + e)},$$

which implies that if $y \geq 1$, we have $h'(y) \geq 0$ and thus $h(y) \geq 0$ for $y \geq 1$. For $y \in [0, 1)$, it is also straightforward to obtain that $h(y) \geq \zeta - \ln(\zeta \ln(1 + e)) \geq 0$. Therefore, we conclude that $h(y) \geq 0$. By substituting $y = C_1$, we have

$$z - (C_0 + C_1 \ln z) \geq 0$$

and thus $x \leq C_0 + C_1 \ln x$ implies $x \leq z$, which finishes the proof. $\qquad\square$

**Lemma G.5** (comparison of measures). *For a sequence $\{a_{t,j}\}$ with $t = 1, ..., T$ and $j = 1, ..., d$, it holds that*

$$\left(\sum_{j=1}^{d} \sqrt{\sum_{t=1}^{T} a_{t,j}^2}\right)^2 \geq \sum_{t=1}^{T} \left(\sum_{j=1}^{d} |a_{t,j}|\right)^2. \tag{34}$$

*Proof.* It holds that

$$\left(\sum_{j=1}^{d} \sqrt{\sum_{t=1}^{T} a_{t,j}^2}\right)^2 \geq \sum_{t=1}^{T} \left(\sum_{j=1}^{d} |a_{t,j}|\right)^2$$

$$\iff \sum_{j_1 \neq j_2}^{d} \sqrt{\sum_{t=1}^{T} a_{t,j_1}^2} \cdot \sqrt{\sum_{t=1}^{T} a_{t,j_2}^2} \geq \sum_{t=1}^{T} \sum_{j_1 \neq j_2}^{d} |a_{t,j_1} a_{t,j_2}|$$

$$\iff \sum_{t=1}^{T} a_{t,j_1}^2 \cdot \sum_{t=1}^{T} a_{t,j_2}^2 \geq \left(\sum_{t=1}^{T} |a_{t,j_1} a_{t,j_2}|\right)^2, \quad \forall j_1 \neq j_2.$$

The last inequality follows from the Cauchy-Schwarz Inequality. Thus we finish the proof. $\qquad\square$

