# OpenReview forum: "AdaGrad under Anisotropic Smoothness"
_ICLR.cc/2025/Conference — ICLR 2025 Poster_

### Official Review · Reviewer_JNL4 · 2024-10-28

**Soundness:** 3
**Presentation:** 3
**Contribution:** 3
**Rating:** 6
**Confidence:** 3

**Summary:**

This paper analyzed the convergence rate of AdaGrad, showing that AdaGrad converges faster than SGD with respect to the dimension of parameters.

**Strengths:**

1. This paper analyzed the convergence rate of AdaGrad. Then, it showed that the convergence rate of SGD depends on $D_{\infty}$, while the rate of AdaGrad depends on $D_2$. $D_2$ depends on the dimension of parameters, while $D_{\infty}$ does not. Thus, this paper claimed that AdaGrad converges faster than SGD.

**Weaknesses:**

1. The authors compared the convergence rate of AdaGrad in Theorem 4.1 and the convergence rate of SGD, Eq. (6). The convergence rate in Eq. (6) depends on $D_2$, while the tighter convergence rate that depends on only $\| x_0 - x^\star\|$ was more common.

2. $D_\infty$ does not depend on the dimension of the parameter. However, it is unclear whether $D_\infty$ is smaller than  $\| x_0 - x^\star\|$.

2. Theorem 4.1 assumes that $L_1 = 0$, which sounds a bit strong assumption. The reviewer feels that it would be better to provide the intuition why this assumption is necessary, at least in the Appendix.

2. It was confusing which term corresponds to "bias term" in line 272.

**Questions:**

See the weakness section.

---

> ### Author Response · Authors · 2024-11-23
> **Response to Reviewer JNL4**
>
> Thank you very much for the careful reading and insightful opinions. Below we would like to provide some discussions on the points.
>
> **Weakness 1 \& 2: SGD convergence can depend on $|| w_0 - w _ * ||_2$.**
>
> Thanks for the insightful question. It is true that the SGD convergence can actually depend on $|| w_0 - w_* ||_2$, but only when the step size is chosen to be $\eta_t \equiv \eta = \mathcal{O}(\frac{1}{\sqrt{T}})$ because we consider a constrained optimization case in convex settings as mentioned in Assumption 3.1. However, it is common and usually more efficient to use a larger step size sequence with $\eta_t = \mathcal{O}(\frac{1}{\sqrt{t}})$ in this setting, which leads to dependence on $D_2$ of the convergence rate (as in Theorem D.1).
>
> **Weakness 3: Theorem 4.1 assumes that $\mathbf{L}_1=0$, which sounds a bit strong assumption.**
>
> Thanks a lot for pointing out. We would like to first apologize for this typo in Theorem 4.1 as $\mathbf{L}_1$ should not appear and we are actually employing Assumption 3.3 here. We didn't use the more general assumption with $\mathbf{L}_1 \neq 0$ in the convex setting mainly because we want to use this simpler convex setting to make it easier for readers to understand how the anisotropic assumptions can help describe the convergence of AdaGrad. Also, for a lot of popular convex machine learning problems, e.g. linear and logistic regression, the smoothness with $\mathbf{L}_1=0$ is sufficient to describe the settings well.
>
> **Weakness 4: It was confusing which term corresponds to "bias term" in line 272.**
>
> Thank you very much for noting the issue. The bias term in line 272 mainly refers to the $\mathcal{O}(1/T)$ term in the convergence rate, which is irrelevant with the noise and corresponds to the convergence of bias in training dynamics. We have made corresponding modifications to make this clearer in our new revision.

---

> ### Author Response · Authors · 2024-11-26
> **Looking forward to further discussion**
>
> Dear Reviewer JNL4,
>
> We deeply appreciate your insightful suggestions and helpful comments. It has been an honor to receive your feedback and to know that you recognize our contribution.
>
> Your assistance in clarifying potential concerns with our manuscript has been invaluable, and we have worked diligently to address them in our response and our new revision. As the discussion period deadline approaches, we would greatly value any further feedback you might have on our response. Your input is crucial in helping us address any remaining concerns.
>
> We fully understand your busy schedule and are sincerely grateful for the effort you have already dedicated to enhancing our work. We look forward to hearing any additional thoughts you may have.
>
> Thank you once again for your guidance and support.
>
> Best,
>
> Submission 8883 authors

---

> ### Comment · Reviewer_JNL4 · 2024-11-27
>
> The reviewer thanks the authors for the response.
>
> > However, it is common and usually more efficient to use a larger step size sequence with $\eta_t = \mathcal{O} (\frac{1}{\sqrt{t}})$ in this setting, which leads to dependence on $D_2$ of the convergence rate (as in Theorem D.1).
>
> The reviewer can not agree with this comment so far. Theorem 5.7 in [1] showed that SGD with $\eta = \frac{1}{4 L \sqrt{t+1}}$ can converge with the rate that does not depend on $D_2$. Please correct me if the reviewer misunderstood.
>
> The reviewer knows that the many existing analyses of the adaptive stepsize, e.g., AdaGrad, also assumed that the set of feasible solutions is bounded, and their convergence rates depend on the diameter of the set of feasible solutions. Thus, the reviewer believes this is a presentation issue and not a critical issue in this paper, but the reviewer feels that it is not fair to compare the rate of SGD shown in Eq. (6) and the rate shown in Theorem 4.1 since we can replace $D_2$ with $\| w_0 - w^\star \|$ in the rate shown in Eq. (6).
>
> Even taking the above points into consideration, the reviewers feel that this paper is overall worthy of being accepted. The reviewer would like to keep the score, which is already positive.
>
> ## Reference
> [1] Garrigos et. al., Handbook of Convergence Theorems for (Stochastic) Gradient Methods, In arXiv 2024

---

> > ### Author Response · Authors · 2024-11-28
> > **Response to Reviewer JNL4**
> >
> > Thank you so much for the insightful feedback and recognition of our paper! We would like to extend our sincere gratitude to the reviewer for identifying this presentation issue, which certainly improve the quality of the paper.
> >
> > You are correct that the convergence rate of SGD can rely on $|| w_0 - w_* || _ 2$ instead of $D_2$ as shown in [1], even for the bounded feasible set setting. This is basically because [1] uses a slightly different proof approach compared to ours (as in Theorem D.1). While this approach can obtain convergence with dependence on $|| w_0 - w_* ||_2$ instead of $D_2$, it suffers from an additional logarithmic term in the bound when using step size $\eta_t = \mathcal{O}\left(\frac{1}{\sqrt{t}}\right)$, as shown in Theorem 5.7 in [1]. Therefore, we think each of these two approaches has its own merits and we take the current approach in our paper for clearer discussion. Many thanks for pointing out this point!
> >
> > We have incorporated relevant modifications and discussions on this problem in Appendix D of the latest revision, which is also mentioned in the main paper. Thank you so much for the constructive suggestion!
> >
> > [1] Garrigos et. al., Handbook of Convergence Theorems for (Stochastic) Gradient Methods, In arXiv 2024

---

### Official Review · Reviewer_14gU · 2024-10-29

**Soundness:** 3
**Presentation:** 4
**Contribution:** 3
**Rating:** 6
**Confidence:** 5

**Summary:**

This paper studies the convergence of Adagrad under anisotropic smoothness conditions. The main contributions are:
1. Defining the anisotropic smoothness condition.
2. Studying the convergence of Adagrad for convex and nonconvex problem under the anisotropic condition.
3. Further nonconvex results for relaxed smoothness conditions.

Strength:

I think the paper is presented in a very clean and organized manner. The key results and discussions are clear.

Up to my knowledge, although anisotropic smoothness were hinted across different setups, there is no very systematic study prior to this work. Therefore, I think the results here can be a valid contribution to optimization theory.

Weakness:

-The results are not surprising, and hence I didn't find the analysis / statements to be novel.


-In addition to reviewing adagrad analyses, it would be helpful to review anisotropic analysis. Several related works that I could think of : analysis on coordinate descent;  Nesterov's study on zero-order methods; adagrad's advantage on sparse gradients; Adam's convergence on infinity norm rather than l2 norm of gradients, etc.

Although, the above results probably are not directly comparable, it would be good to summarize and discuss the differences.

Some results that can make the work more impressive are listed below:

-Lower bounds to justify when and why adaptive step,  diagonal / full matrix adaptivity are sufficient / insufficient would be very interesting.

-Given the analysis, can faster methods be proposed for neural nets?

**Strengths:**

See summary

**Weaknesses:**

See summary

**Questions:**

See summary

---

> ### Author Response · Authors · 2024-11-23
> **Response to Reviewer 14gU**
>
> We would like to express our appreciation for the reviewer's effort in checking our paper. Thanks! We are happy to provide the following discussions regarding the mentioned points.
>
> **Weakness 1: The results are not surprising, and hence I didn't find the analysis/statements to be novel.**
>
> Thanks for the comment. We would like to also make some further clarifications of the novelty of this work:
>
> 1. To our knowledge, this result is the first one that theoretically shows the benefits of coordinate-wise step size in the general smooth optimization settings.
> The result is as expected based on numerous empirical observations, but being the first one to concretely lay the foundation to bridge theory and practice is also important, which can shed light on a more rigorous understanding of the effectiveness of adaptive gradient methods and inspire even more efficient algorithms in the future.
>
> 2. Regarding the technical novelty, as we have discussed in Remark 5.4, our analysis refers to the main line of existing results but incorporates better proof techniques to enable more general assumptions and clean results.
>
> **Weakness 2: In addition to reviewing Adagrad analyses, it would be helpful to review anisotropic analysis.**
>
> This is certainly a good suggestion! Beyond AdaGrad, we have also discussed sign-based optimization methods that uses anisotropic smoothness assumptions, existing understanding of advantages of adaptive gradient methods in Section 2. In our new revision, we additionally add some more discussions on coordinate descent and other related work might employ the anisotropic smoothness assumption.
>
> **Weakness 3: Some results that can make the work more impressive.**
>
> Thanks a lot for the insightful suggestions. A lower bound for the adaptivity over dimensions can be of interest to see whether adaptive gradient methods are good enough and whether we can, as the reviewer pointed out, design even faster algorithms.
>
> However, as these investigations might be beyond our topic in this paper, we will carefully consider these topics as possible future directions. Thanks again for the suggestion!

---

> > ### Comment · Reviewer_14gU · 2024-11-26
> > **Thanks**
> >
> > I thank the authors for the response.
> >
> > I would like to keep my score, which is already positive.

---

> > > ### Author Response · Authors · 2024-11-26
> > >
> > > Thanks so much for the insightful suggestions and recognizing the contribution of our paper! We really appreciate it.

---

### Official Review · Reviewer_zDPR · 2024-11-01

**Soundness:** 4
**Presentation:** 3
**Contribution:** 3
**Rating:** 8
**Confidence:** 4

**Summary:**

The work provide analysis result for the convergence of Adagrad for training of machine learning models with large batch size, emphasizing the effects of anisotropic smoothness. The authors then compare the results with similar results for SGD and Adagrad-norm and point out the potential of Adagrad. In general, the work can be helpful to under stand the training process.

**Strengths:**

The work provides an analysis result which may be the first one for Adagrad. This can be helpful for others to understand the potential of Adagrad and select optimizers for training tasks.

**Weaknesses:**

The numerical results are not sufficient to verify the assumptions and analytic results.

**Questions:**

1. Convexity is used in a large part of the paper. In many machine learning models, there are more parameters than data. In this case, local minima w* may not be isolated points. Instead, it can be a manifold. What is the impact of overparametrization to the results in this work?

2. In Table 2, the authors list the coefficients and norms in the analytical results. It is also important to see how well the convergence of the loss (or gradients) are controlled by these coefficients and norms.

3. For Table 4, since the work mainly discusses the convergence rate of Adagrad, it is better to show how the loss converges and how do the  authors select hyperparameters.

---

> ### Author Response · Authors · 2024-11-23
> **Response to Reviewer zDPR**
>
> We would like to extend our sincere appreciation to the reviewer for all the constructive and positive feedback on our contribution. Many thanks!
>
> **Question 1: What is the impact of overparametrization to the results in this work?**
>
> Thanks for the insightful suggestion! We add a part in the paper discussing the impact of over-parameterization in convex settings in Appendix E.1. Basically, we consider the noise $\sigma_*$ only on $w_*$, the optimum manifold. In this case, we change the anisotropic smoothness to be expected anisotropic smoothness and obtain convergence results under these adjusted assumptions. By this setting, we have the following convergence rate for over-parameterized models (which usually result in interpolation of the data and thus $\sigma_* = E[|\nabla_{w} f(w_*;\xi)|] = 0$):
>     $$
>     \mathcal{O}\left( \frac{||L_*|| _ 1 D_\infty^2}{T} \right)
>     $$
> where $L_* \in \mathbb{R}^d$ is the anisotropic expected smoothness. Therefore, we basically improve the convergence rate to $\mathcal{O}(1/T)$ for over-parameterized models.
>
> **Question 2: It is also important to see how well the convergence of the loss (or gradients) are controlled by these coefficients and norms.**
>
> Thanks for the constructive suggestion. We will definitely include the mentioned results in our final revision.
>
> **Question 3: It is better to show how the loss converges and how do the authors select hyperparameters.**
>
> Thanks for the helpful comment. We include the corresponding results in Table 3 and Figure 2 for loss convergence and Appendix A for the parameter selection.

---

### Official Review · Reviewer_FCcu · 2024-11-04

**Soundness:** 3
**Presentation:** 3
**Contribution:** 3
**Rating:** 6
**Confidence:** 3

**Summary:**

This paper provides a detailed analysis of the AdaGrad optimization algorithm under anisotropic smoothness assumptions, addressing gaps in theoretical convergence for large-scale tasks. It introduces a new anisotropic smoothness framework that better explains AdaGrad’s convergence speed, especially for large-batch training. Experiments on logistic regression and GPT-2 fine-tuning support these theoretical claims, showing AdaGrad’s improved performance over SGD.

**Strengths:**

The main strength lies in its novel anisotropic assumptions, which align well with AdaGrad’s observed performance in high-dimensional settings. The experiments effectively validate the theoretical benefits, highlighting AdaGrad’s adaptability to large batch sizes and diverse data structures. For the rest it is a standard optimization analysis.

**Weaknesses:**

This kind of work always relies on assumptions which limits their applicability to the setting of interests, as neural networks. However, this is common and not really an issue. See also questions.

**Questions:**

Can please you better compare to Convergence Analysis of Adaptive Gradient Methods under Refined Smoothness and Noise Assumptions - D Maladkar, R Jiang, A Mokhtari - arXiv preprint arXiv:2406.04592, 2024?
Also, is much work required to generalize to Adam?

---

> ### Author Response · Authors · 2024-11-23
> **Response to Reviewer FCcu**
>
> We thank the reviewer for the insightful comments and questions. We would like to provide the following discussions regarding the weakness part and questions.
>
> **Weakness: Assumptions limits the applicability to the setting of interests, as neural networks.**
>
> We appreciate the reviewer's idea that theoretical assumptions can limit the application of the theory. But we also want to kindly note that our key novel assumption, the anisotropic generalized smoothness assumption has been verified by GPT-2 experiments in Figure 1, which provide a certain degree of evidence for bridging the theory and practice in realistic settings like training neural networks.
>
> **Question: Better Comparison with (Maladkar et. al. 2024).**
>
> Thanks for the question. The starting point of this work and Maladkar et. al. (2024) is quite similar in that we both utilize anisotropic smoothness assumptions to obtain better convergence results for AdaGrad. However, there are some major differences between the two papers.
>
> 1. As we noted in the paper, our results apply to the anisotropic generalized smoothness settings. This is a more general setting that can well describe practical neural network training (as shown in Figure 1) and covers the results in Maladkar et. al. (2024).
>
> 2. We include a convex part in our paper, which is a simpler case for illustrating the power of anisotropic assumptions. Based on this part and our new assumption and theory, we also did various empirical verifications in the experiment part, which provides strong evidence for our theory.
>
> 3. We also note that compared to ours, Maladkar et. al. (2024) introduced a lower bound for gradient $\ell_1$-norm convergence of SGD under nonconvex anisotropic assumptions, which may help a more rigorous comparison between SGD and AdaGrad in the nonconvex case. However, as we have discussed in Section 4, it's not clear whether $\ell_1$-norm is a better convergence measure and it actually depends on the properties of the training curvature.
>
> To conclude, both the work contribute to a better theoretical understanding of the benefits of adaptive gradient methods.
>
> We have also included this more detailed discussion in our revision. Thanks again for the constructive suggestion.
>
> **Question: Is much work required to generalize to Adam?**
>
> We believe generalization to Adam should be applicable, but technical issues remain. Basically the main problem is how to deal with the momentum together with the adaptive gradient preconditioner in the stochastic nonconvex cases, which makes most Adam analysis worse than AdaGrad ones, in terms of worse convergence rates and/or stricter noise assumptions. Because of the failure to prove the benefits of adding the momentum term, currently, we think an extension to Adam might only provide little insight. However, this should definitely be a good future direction.

---

> ### Author Response · Authors · 2024-11-26
> **Looking forward to further discussions**
>
> Dear Reviewer FCcu,
>
> We deeply appreciate your insightful suggestions and helpful comments. It has been an honor to receive your feedback and to know that you recognize our contribution.
>
> Your assistance in making a more detailed related work and discussion on further extensions has been invaluable, and we have worked diligently to address this in our response and our new revision. As the discussion period deadline approaches, we would greatly value any further feedback you might have on our response. Your input is crucial in helping us address any remaining concerns.
>
> We fully understand your busy schedule and are sincerely grateful for the effort you have already dedicated to enhancing our work. We look forward to hearing any additional thoughts you may have.
>
> Thank you once again for your guidance and support.
>
>
> Best,
>
> Submission 8883 authors

---

> > ### Comment · Reviewer_FCcu · 2024-11-27
> >
> > Thanks for the clarifications! I appreciate that you incorporated more comments about it. I will update my grade to 6.

---

> > > ### Author Response · Authors · 2024-11-28
> > > **Response to Reviewer FCcu**
> > >
> > > Thank you so much for your recognition of our paper. We appreciate the time and effort you dedicated in reviewing our paper. Your suggestions have been invaluable in improving our manuscript!

---

### Meta-Review · Area_Chair_rZ5t · 2024-12-10

**Metareview:**

This paper studies AdaGrad under anisotropic smoothness conditions, addressing its theoretical convergence and comparing its performance with SGD. The reviewers largely appreciated the theoretical analysis and the new condition introduced in the paper.

### **Strengths:**
- Introduces anisotropic smoothness to explain AdaGrad's convergence speed.
- Provides convergence results under both convex and nonconvex settings, with comparisons to SGD and AdaGrad-Norm.
- Highlights AdaGrad's adaptability to large batch sizes  which is backed up by experiments

### **Weaknesses:**
- Initial experiments were limited, and reviewers requested more comprehensive validation, particularly on real-world neural networks and parameter selection.
- Comparison to related studies (e.g., Maladkar et al., 2024; Garrigos et al., 2024) could have been more explicit, particularly in terms of convergence rates and dimensionality dependencies.
- Some reviewers suggested generalizing the framework to other optimizers like Adam

Overall, the positive reviews make the paper suitable for acceptance.

**Additional Comments On Reviewer Discussion:**

There was some discussions that lead the reviewers to have a more positive opinion of the paper

---

### Decision · Program_Chairs · 2025-01-22

Accept (Poster)